# DV-3DLane: End-to-end Multi-modal 3D Lane Detection with Dual-view Representation

**Yueru Luo[1,2], Shuguang Cui[2,1], Zhen Li[2,1]***
[1] FNii, CUHK-Shenzhen    [2] School of Science and Engineering, CUHK-Shenzhen
{222010057@link.,shuguangcui@,lizhen@}cuhk.edu.cn

## ABSTRACT

Accurate 3D lane estimation is crucial for ensuring safety in autonomous driving. However, prevailing monocular techniques suffer from depth loss and lighting variations, hampering accurate 3D lane detection. In contrast, LiDAR points offer geometric cues and enable precise localization. In this paper, we present DV-3DLane, a novel end-to-end **D**ual-**V**iew multi-modal **3D Lane** detection framework that synergizes the strengths of both images and LiDAR points. We propose to learn multi-modal features in dual-view spaces, *i.e.*, *perspective view* (PV) and *bird's-eye-view* (BEV), effectively leveraging the modal-specific information. To achieve this, we introduce three designs: **1)** A bidirectional feature fusion strategy that integrates multi-modal features into each view space, exploiting their unique strengths. **2)** A unified query generation approach that leverages lane-aware knowledge from both PV and BEV spaces to generate queries. **3)** A 3D dual-view deformable attention mechanism, which aggregates discriminative features from both PV and BEV spaces into queries for accurate 3D lane detection. Extensive experiments on the public benchmark, OpenLane, demonstrate the efficacy and efficiency of DV-3DLane. It achieves state-of-the-art performance. with a remarkable **11.2** gain in F1 score and a substantial **53.5%** reduction in errors. The code is available at https://github.com/JMoonr/dv-3dlane.

## 1 INTRODUCTION

Autonomous driving (AD) technology in recent years has made remarkable strides, bringing us closer to the realization of fully self-driving vehicles. Within this field, one of the key challenges is the accurate detection of 3D lanes, a critical component for ensuring safe and reliable navigation. 3D lane detection entails identifying the 3D positions of lane boundaries in the environment, providing essential data for tasks like path planning and vehicle control.

3D lane detection is proposed to mitigate the limitations posed by the absence of depth information in 2D prediction. Currently, the majority of 3D lane detection methods rely on vision-centric approaches, *i.e.*, monocular solutions, where some designs are naturally borrowed and benefit from advances in 2D lane methods. Taking the perspective-view (PV) image as input,

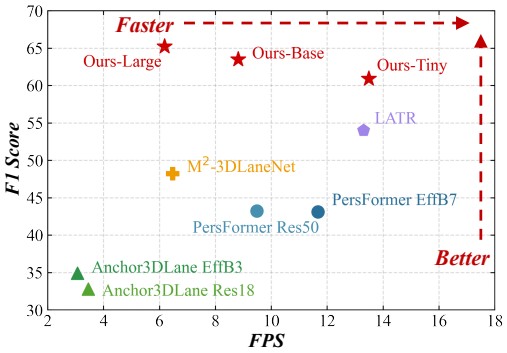

Figure 1: **FPS vs. F1 score.** All models are tested on a single V100 GPU, and F1-score is evaluated with a harsh distance threshold of **0.5m** on the OpenLane-1K dataset. Our model sets a new state-of-the-art, and our tiny version surpasses all previous methods with the fastest FPS. More details can be found in Table 1 and our Appendix.

these monocular methods mainly utilize the inverse perspective mapping (IPM) Mallot et al. (1991) technique to warp the PV features into BEV. However, there are misalignment issues in the IPM-based methods when encountering non-flat roads, due to the rigid flat assumption of IPM Nedevschi et al. (2004); Yan et al. (2022). While some recent efforts have been made to address this issue and

---

*Corresponding author.

have shown promising results by directly predicting 3D lanes in PV Bai et al. (2022b); Huang et al. (2023); Luo et al. (2023), these monocular 3D approaches, as vision-centric solutions, inevitably get stuck in capturing the complexity of real-world driving scenarios, when encountering adverse weather and lighting conditions. In contrast, as an active sensor, LiDAR excels in spatial localization and 3D structure perception, complementing the capabilities of passive sensor cameras, and it gets more widely used thanks to hardware advancements. A bunch of recent works in 3D object detection have demonstrated the power of LiDARs Zhou & Tuzel (2018); Lang et al. (2019); Yin et al. (2021a) and multiple modalities Liang et al. (2019); Wang et al. (2021); Yang et al. (2022); Li et al. (2022b); Chen et al. (2023) in autonomous driving scenarios. Whereas, *fewer endeavors* Bai et al. (2018); Luo et al. (2022) *have been made to exploit multi-modal strength for 3D lane detection*. Albeit using extra LiDAR data, $M^2$-3DLane Luo et al. (2022) failed to make full use of features in image space which is crucial to 3D lane performance. Besides, $M^2$-3DLane employs a naive fusion to aggregate multi-modal features, resulting in inferior performance to the camera-only methods(*e.g.*, Luo et al. (2023)).

Given the rich semantics inherent in images and the accurate positional information afforded by the BEV representation Philion & Fidler (2020); Li et al. (2022d), we strive to exploit the multi-modal features to enhance the performance of 3D lane detection. Existing methods tend to fuse two modalities into a *single* space Liang et al. (2022); Liu et al. (2023b), *e.g.*, BEV, for feature extraction and subsequent prediction. However, this approach constrains the model's capacity to harness modality-specific features. We contend that features represented in *both PV space and BEV space bear significance*, facilitating improved representation learning. Motivated by the above observation, we introduce **DV-3DLane**, a novel end-to-end multi-modal 3D lane detection framework.

To maintain a dual-view space representation, we adopt a symmetric backbone consisting of a PV branch and a BEV branch to extract features in PV and BEV spaces, respectively. To leverage the merits of both images and points for comprehensive feature learning in each view, we design a *bidirectional* feature fusion (BFF) strategy. Subsequently, to effectively facilitate query-based detection using the retained dual-view features, we devise a *unified* query generator (UQG). This generator initially produces two sets of lane-aware queries: one from the PV space and the other from the BEV space. These two query sets are compelled to capture lane knowledge regarding semantics and spatiality, guided by auxiliary 2D segmentation supervision. Further, these two sets are then combined into a unified set that serves the decoder. To achieve the unification of dual-view queries, we propose a *lane-centric* clustering technique. Besides, we employ a Transformer decoder to aggressively integrate discriminative features from both views into the unified queries. For effective feature aggregation across different view spaces, we introduce a 3D dual-view deformable attention mechanism that considers the inherent properties of 3D space, resulting in deformed 3D sample points. These 3D sample points are then projected onto the PV and BEV planes, yielding 2D sample points in each respective view space. These projected 2D points are utilized for feature sampling within their respective view spaces.

In summary, our contributions are threefold :

- We introduce DV-3DLane, an end-to-end multi-modal 3D lane detection framework that harnesses the power of dual-view representation.

- We devise the BFF strategy to mutually fuse features across modalities, and design the UQG to merge lane-aware queries from dual views, yielding a unified query set. Further, a 3D dual-view deformation attention mechanism is introduced to aggregate dual-view features effectively.

- We conduct thorough experiments on the OpenLane benchmark to validate the effectiveness of our method. Experimental results show that DV-3DLane surpasses previous methods significantly, achieving an impressive **11.2 gain** in F1 score and a remarkable **53.5% reduction** in errors. Moreover, a 3D dual-view deformation attention mechanism is introduced to aggregate dual-view features effectively.

## 2 RELATED WORK

### 2.1 2D LANE DETECTION

Recent works in 2D lane detection can be broadly categorized into four main approaches: **1)** Segmentation-based methods Lee et al. (2017); Pan et al. (2017); Neven et al. (2018); Hou et al.

(2019); Xu et al. (2020); Zheng et al. (2021) devote to classifying pixels into lanes or the background, necessitating further post-processing steps (*e.g.*, grouping and curve fitting) to produce lane instances. **2)** Anchor-based methods, inspired by region-based object detectors such as Faster-RCNN Ren et al. (2015), employ line-like anchors to localize lanes Wang et al. (2018); Li et al. (2019); Tabelini et al. (2021a). To overcome the limitations of straight-line constraints, Jin et al. (2022) employ eigenlane space to produce diverse lane shape candidates. **3)** Point-based methods Ko et al. (2021); Qu et al. (2021); Wang et al. (2022); Xu et al. (2022) attempt to flexibly localize key points along each lane instance and subsequently group the points belonging to the same lane. **4)** Parametric methods Van Gansbeke et al. (2019); Tabelini et al. (2021b); Liu et al. (2021); Feng et al. (2022) formulate lane detection as a curve fitting problem, leveraging prior knowledge about lane shapes by representing them using various parametric forms, such as polynomials and splines.

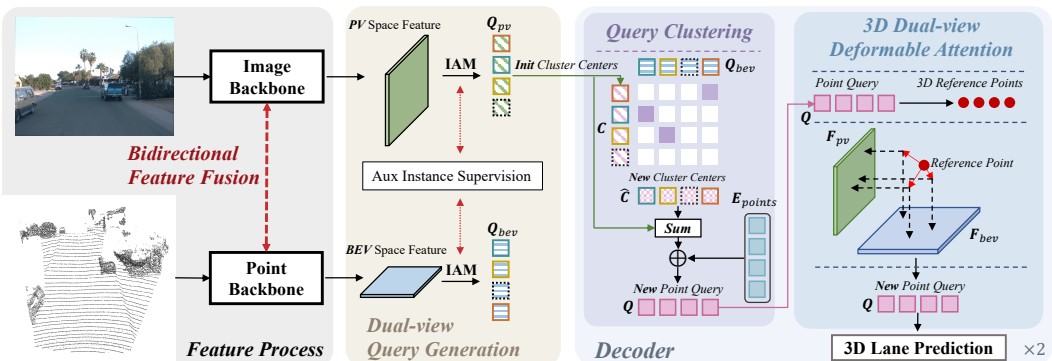

Figure 2: **Overview of DV-3DLane.** First, images and point clouds undergo separate processing by the image backbone and point backbone. In the middle stage of backbones, we introduce Bidirectional Feature Fusion (BFF) to fuse multi-modal features across views. Subsequently, the instance activation map (IAM) is utilized to produce lane-aware queries $\mathbf{Q}_{pv}$ and $\mathbf{Q}_{bev}$. These queries are then subjected to Dual-view Query Clustering, which aggregates dual-view query sets $\mathbf{Q}_{pv}$ and $\mathbf{Q}_{bev}$ into a unified query set $\mathbf{C}$, further augmented with learnable point embeddings $\mathbf{E}_{points}$ to form query $\mathbf{Q}$. Additionally, we introduce 3D Dual-view Deformable Attention to consistently aggregate point features from both view features $\mathbf{F}_{pv}$ and $\mathbf{F}_{bev}$ into $\mathbf{Q}$. $\oplus$ denotes broadcast summation. Notably, the $\oplus \mathbf{E}_{points}$ operation is performed only in the first layer, while in the following layer, $\oplus \mathbf{Q}$ is utilized. Different colored boxes □□□ denote queries targeting different lanes; dashed boxes ⬚ represent the background, and box texture indicates features.

## 2.2 3D LANE DETECTION

Existing methods center on vision-centric solutions and draw inspiration from the 2D task. Typically, monocular approaches Garnett et al. (2019); Efrat et al. (2020); Guo et al. (2020); Chen et al. (2022); Wang et al. (2023); Liu et al. (2022); Li et al. (2022a); Ai et al. (2023); Yao et al. (2023) construct surrogate representations using inverse perspective mapping (IPM), and perform predictions in this surrogate space. Nonetheless, IPM inherently introduces discrepancies between the perspective and the surrogate view in non-flat areas due to its planar assumption. To address this limitation, recent efforts have endeavored to predict 3D lanes from the perspective view Yan et al. (2022); Bai et al. (2022b); Huang et al. (2023); Luo et al. (2023), or employ a depth-aware projection to enhance lane perception by incorporating LiDAR information Luo et al. (2022).

## 2.3 MULTI-MODAL DETECTION

Despite advancements in lane detection, multi-modal methods remain relatively underexplored. Previous works typically utilize either BEV Bai et al. (2018); Yin et al. (2020); Luo et al. (2022) or PV Zhang et al. (2021b) as representation spaces for performing 2D lane segmentation Yin et al. (2020); Zhang et al. (2021b) or 3D lane detection Bai et al. (2018); Luo et al. (2022). For BEV-based methods, Bai et al. (2018) rasterizes LiDAR points to create a BEV image and transforms PV images into BEV using the estimated ground height derived from the LiDAR data. Similarly, M$^2$-3DLane Luo et al. (2022) utilizes the BEV space to fuse multi-modal features. To project PV features into BEV space, they lift compact 2D features into 3D space guided by the depth map and

further employ a pillar-based method Lang et al. (2019) to splat them into BEV. While these methods primarily focus on 3D tasks, Yin et al. (2020) leverages BEV space for fusing camera and LiDAR features, serving for 2D BEV lane segmentation. Conversely, Zhang et al. (2021b) adopts PV to fuse multi-modal features for 2D lane segmentation. In contrast to lane detection, multi-modal methods have been extensively studied in 3D object detection, with most previous multi-modal methods attempting to fuse image features into BEV space due to its compactness and interoperability for ambient perception Ma et al. (2022). These methods either adopt point-level fusion Sindagi et al. (2019); Wang et al. (2021); Yin et al. (2021b) to paint points, instance-level fusion to project 3D proposals to image space Yoo et al. (2020); Bai et al. (2022a), or feature-level fusion to transform features from PV space into BEV space Liu et al. (2023b); Liang et al. (2022). However, few works consider both the perspective view and BEV simultaneously.

## 3 METHODOLOGY

The overall framework of our DV-3DLane is depicted in Figure 2. Section 3.1 describes the bidirectional feature fusion module, which merges different modalities bidirectionally and constructs multi-modal features in both PV and BEV spaces. In Section 3.2, we present the unified query generator, which generates two lane-aware query sets from dual views and unifies them into a shared space in a lane-centric manner. Section 3.3 introduces the 3D dual-view deformable attention module, which effectively aggregates dual-view features into unified queries, serving for prediction.

### 3.1 BIDIRECTIONAL FEATURE FUSION

Instead of merging different views into one single space Bai et al. (2018); Luo et al. (2022); Liang et al. (2022); Li et al. (2022d); Liu et al. (2023b), we propose to retain features in both PV and BEV spaces while incorporating multi-modal features for each view. To achieve this, we employ a dual branch to extract features for each view, using images and points as input, respectively. Intermediately, we conduct *bidirectional* feature fusion between the symmetric branches to enhance each view with multiple modalities, as shown in Figure 3 and summarized in Algorithm 1.

---

**Algorithm 1** Bidirectional Feature Fusion (BFF)

**Input**: LiDAR points $\mathbf{P_{pt}}$, image $\mathbf{I}$, camera parameters $\boldsymbol{T}$
**Output**: mm-aware PV features $\mathbf{F}_{pv}$, BEV features $\mathbf{F_{bev}}$,
*"mm" denotes multi-modal.*

$\quad \mathbf{F}_{pt}^{s1} = \text{PillarNet-S1}(\mathbf{P}_{pt}), \mathbf{F}_{pv}^{s1} = \text{ResNet-S1}(\mathbf{I})$
$\qquad\qquad\qquad\qquad\qquad\qquad\qquad \triangleright \textit{S1: stage one.}$
$\quad \mathbf{P}_{pt2pv} = \{(u_i, v_i)|i \in P\} = \text{Project}(\boldsymbol{T}, \mathbf{P}_{pt})$
$\quad \mathbf{F}_{pt2pv} = \text{Scatter}(idx = \mathbf{P}_{pt2pv}\ src = \mathbf{F}_{pt}^{s1})$
$\qquad\qquad\qquad\qquad\qquad\qquad \triangleright \textit{points} \rightarrow \textit{pixels.}$
$\quad \mathbf{F}_{pv2pt} = \text{Grid\_Sample}(src = \mathbf{F}_{pv}^{s1}, coords = \mathbf{P}_{pt2pv})$
$\qquad\qquad\qquad\qquad\qquad\qquad \triangleright \textit{pixels} \rightarrow \textit{points.}$
$\quad \mathbf{F}_{pv} = \text{ResNet}(\text{Concat}(\mathbf{F}_{pv}^{s1}, \mathbf{F}_{pt2pv}))$
$\quad \mathbf{F}_{bev} = \text{PillarNet}(\text{Concat}(\mathbf{F}_{pt}^{s1}, \mathbf{F}_{pv2pt}))$
$\qquad\qquad\qquad \triangleright \textit{dual-view multi-modal feature extraction.}$

---

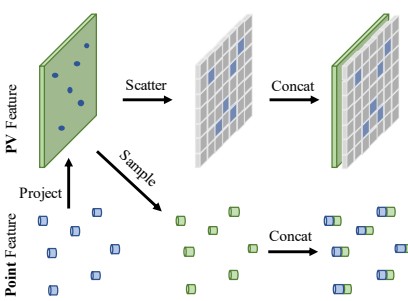

Figure 3: **Bidirectional Feature Fusion (BFF).** We represent the image feature in green and points in blue.

Concretely, we place points and images in their designated branches. After obtaining low-level features within each branch, we perform bidirectional feature fusion. By projecting 3D points $\mathbf{P}_{pt} = \{(x_i, y_i, z_i)|i \in P\}$ onto the PV plane, we obtain their corresponding 2D coordinates $\mathbf{P}_{pt2pv} = \{(u_i, v_i)|i \in P\}$, where $P$ is the cardinality of the point set. **1)** For *points-to-pixels* fusion, we utilize a Scatter operation to construct dense point feature grids $\mathbf{F}_{pt2pv}$, (depicted in the upper part of Figure 3, with blue cells denoting positions hit by the projected 3D points). **2)** For *pixels-to-points* fusion, we employ bilinear interpolation to sample features at 2D positions hit by the projection of 3D points, yielding $\mathbf{F}_{pv2pt}$ (shown in the lower part of Figure 3). The resulting cross-modal features in PV and BEV are concatenated with their respective original modal features. The fused multi-modal features in each view, *i.e.*, PV and BEV, are then fed into subsequent modules in the corresponding branch, generating $\mathbf{F}_{pv}$ and $\mathbf{F}_{bev}$, respectively. Notably, $\mathbf{F}_{pv}$ and $\mathbf{F}_{bev}$ encapsulate multi-modal information represented in distinct spaces.

## 3.2 UNIFIED QUERY GENERATOR

We introduce a unified query generator for end-to-end 3D lane detection. To this end, we first generate two distinct lane-aware query sets, termed dual-view queries, from the previously obtained multi-modal features, $\mathbf{F}_{pv}$ and $\mathbf{F}_{bev}$. Then, we present a lane-centric clustering strategy to unify these dual-view queries into a cohesive set of queries.

**Dual-view Query Generation.** To effectively capture semantic and spatial features related to lanes, which are termed as "lane-aware" knowledge, we utilize an instance activation map (IAM) Cheng et al. (2022)-assisted method to generate lane-aware queries in PV and BEV spaces. Taking PV branch as an example, we produce a set of IAMs, denoted as $\mathbf{A}_{pv}$, via the following equation:

$$\mathbf{A}_{pv} = \sigma(\mathcal{F}(\text{Concat}(\mathbf{F}_{pv}, \mathbf{S}_{pv}))),$$

where $\mathbf{A}_{pv} \in \mathbb{R}^{N \times H_{pv} \times W_{pv}}$, $\mathbf{F}_{pv} \in \mathbb{R}^{C \times H_{pv} \times W_{pv}}$, $N$ denotes query number, $\sigma$ is the sigmoid function, Concat represents concatenation operation, and $\mathbf{S}_{pv}$ comprises two-channel spatial localization features for each pixel Liu et al. (2018). The lane-aware query $\mathbf{Q}_{pv}$ assisted by IAMs is generated via:

$$\mathbf{Q}_{pv} = \mathbf{A}_{pv} \otimes \mathbf{F}_{pv}^{\mathsf{T}},$$

where $\mathbf{Q}_{pv} \in \mathbb{R}^{N \times C}$, $\otimes$ denotes the matrix product. Similarly, lane-aware BEV query $\mathbf{Q}_{bev} \in \mathbb{R}^{N \times C}$ is formed using:

$$\mathbf{Q}_{bev} = \sigma(\mathcal{F}([\mathbf{F}_{bev}, \mathbf{S}_{bev}])) \otimes \mathbf{F}_{bev}^{\mathsf{T}}.$$

To force the query sets to learn lane-aware features, during training, we employ an auxiliary instance segmentation for each branch on top of the query set. Labels for the auxiliary segmentation are generated in pairs for these two branches, which are further assigned to predictions using mask-based bipartite matching Cheng et al. (2022), as illustrated in Figure 4 (a) and (b).

**Dual-view Query Clustering.** Given dual-view query sets $\mathbf{Q}_{pv}$ and $\mathbf{Q}_{bev}$, we propose employing a *lane-centric* clustering technique to generate a unified query set for end-to-end lane detection. While *k*Max-DeepLab Yu et al. (2022) previously used k-means cross-attention to group pixels into distinct clusters, *i.e.*, instance masks, our approach focuses on unifying queries from different views. Queries from $\mathbf{Q}_{pv}$ and $\mathbf{Q}_{bev}$ targeting the same lane are merged within the same cluster. Specifically, we initiate lane cluster centers $\mathbf{C} \in \mathbb{R}^{N \times C}$ with $\mathbf{Q}_{pv}$,

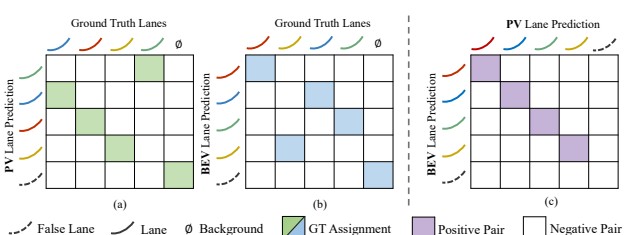

Figure 4: Illustration of one-to-one matching and lane-centric clustering. (a) and (b) show the assignment for BEV and PV predictions, respectively. (c) depicts the pairing of the clustering, where queries targeting the same lane are treated as a positive pair, otherwise negative.

and assign each query in $\mathbf{Q}_{bev}$ to its nearest cluster center among $\mathbf{C}$. Notably, cluster centers can be chosen from either $\mathbf{Q}_{pv}$ or $\mathbf{Q}_{bev}$. Empirically, we found that using $\mathbf{Q}_{pv}$ produces better results. To achieve clustering, we perform attention between $\mathbf{C}$ (query) and $\mathbf{Q}_{bev}$ (key), while applying $\text{argmax}$ along the cluster center (query) dimension Yu et al. (2022) as follows:

$$\mathbf{A} = \underset{N}{\text{argmax}}(\mathbf{C} \times \mathbf{Q}_{bev}^{\mathsf{T}}), \quad \hat{\mathbf{C}} = \mathbf{A} \cdot \mathbf{Q}_{bev} + \mathbf{C},$$

where $\hat{\mathbf{C}} \in \mathbb{R}^{N \times C}$ refers to updated centers unifying queries from dual views. In practise, we use gumbel-softmax Jang et al. (2016); Liang et al. (2023) to substitute $\text{argmax}$.

Considering the variation and slenderness of lanes, we employ a refined point query scheme Luo et al. (2023) to enhance lane detection. Instead of using a single query for each lane, multiple-point queries are employed for more precise capture Luo et al. (2023); Liao et al. (2022); Zhang et al. (2021a); Liu et al. (2023a). Consequently, in the first layer, we construct point-based queries $\mathbf{Q} \in \mathbb{R}^{N \times M \times C}$ with $\mathbf{Q} = \hat{\mathbf{C}} \oplus \mathbf{E}_{points}$, where $\oplus$ denotes broadcast sum, $\mathbf{E}_{points} \in \mathbb{R}^{M \times C}$ is the learnable point embedding, and in the subsequent layer, we update $\mathbf{Q}$ by $\mathbf{Q} = \hat{\mathbf{C}} \oplus \mathbf{Q}$.

**Supervision on Query Clustering.** Given the critical importance of deep supervision for the clustering Yu et al. (2022), we leverage the InfoNCE loss Oord et al. (2018) to supervise the query

clustering in a lane-centric manner, as illustrated in Figure 4 (c) and formulated as:

$$\mathcal{L}_{\text{NCE}} = -\log \frac{\exp(\boldsymbol{q} \cdot \boldsymbol{k}^+/\tau)}{\exp(\boldsymbol{q} \cdot \boldsymbol{k}^+/\tau) + \sum_{\boldsymbol{k}^- \in \mathcal{N}} \exp(\boldsymbol{q} \cdot \boldsymbol{k}^-/\tau)},$$

where $\tau$ is a temperature hyper-parameter Wu et al. (2018), $\boldsymbol{q}$ denotes one query, $\boldsymbol{k}^+$ indicates the positive sample *w.r.t.* $\boldsymbol{q}$, and $\mathcal{N}$ denotes the collection of all negative samples from the different query set relative to the one containing $\boldsymbol{q}$. Notably, queries assigned to the background do not incur penalties in the clustering learning process. With this supervision, queries from different views are grouped together when matched to the same ground truth lane. Consequently, lane-aware knowledge residing in two view spaces is synergized into the unified query.

## 3.3 3D DUAL-VIEW DEFORMABLE ATTENTION

Apart from informative query generation, feature aggregation plays a crucial role in DV-3DLane. Instead of projecting points from densely sampled grids Chen et al. (2022) or their lifted pillars Li et al. (2022d) onto the PV plane for feature sampling, as shown in Figure 5 (a), we adopt sparse queries to sample features from different views. Moreover, our approach distinguishes itself from several existing sparse query methods, as depicted in Figure 5 (b) and (c). For instance, DeepIntera-tion Yang et al. (2022) (Figure 5 (b)) employs a sequential method to sample PV and BEV features, while FUTR3D Chen et al. (2023) (Figure 5 (c)) projects 3D points into different spaces, sampling features individually for each space.

In contrast, as outlined in Algorithm 2, we leverage the inherent properties of 3D space by predicting both 3D reference points and their 3D offsets using queries, forming 3D deformed points. These 3D deformed points are then projected into each space, establishing a *consistent* feature sampling strategy across spaces, as depicted in Figure 5. Consequently, features corresponding to the same 3D points from different views are effectively sampled and integrated into the query.

---

**Algorithm 2** 3D DV Deformable Attention

**Input**: unified query set $\mathbf{Q}$, PV features $\mathbf{F}_{pv}$, BEV features $\mathbf{F}_{bev}$, camera parameters T.
**Output**: updated unified query $\mathbf{Q}$.

$\quad\mathbf{Ref}_{3d} = \text{MLP}_1(\mathbf{Q})$
$\qquad\qquad\qquad\qquad\qquad \triangleright$ *3D reference points.*
$\quad\boldsymbol{\Delta}\mathbf{Ref}_{3d} = \text{MLP}_2(\mathbf{Q})$
$\quad\mathbf{S}_{3d} = \{(x_i, y_i, z_i)|i \in N\} = \boldsymbol{\Delta}\mathbf{Ref}_{3d} + \mathbf{Ref}_{3d}$
$\qquad\qquad\qquad\qquad\qquad \triangleright$ *deformed 3D positions.*
$\quad\mathbf{D}_{pv} = \text{DeformAttn}(\text{Project}_{pv}(\mathbf{S}_{3d}, \text{T}), \mathbf{F}_{pv})$
$\qquad\qquad\qquad \triangleright$ *project 3D deformed points to PV.*
$\quad\mathbf{D}_{bev} = \text{DeformAttn}(\text{Project}_{bev}(\mathbf{S}_{3d}), \mathbf{F}_{bev})$
$\quad\mathbf{Q} = \text{SE}(\mathbf{D}_{pv}, \mathbf{D}_{bev})$

---

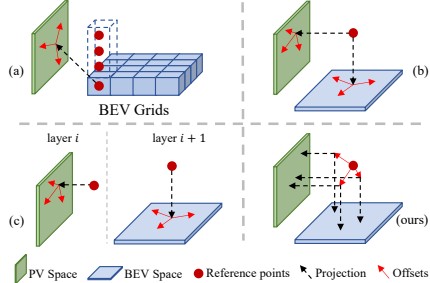

Figure 5: Illustration comparing 3D dual-view deformable attention with other approaches.

## 3.4 PREDICTION AND LOSS

**Auxiliary Tasks.** During training, we incorporate two auxiliary tasks: 1) 2D instance segmenta-tion Luo et al. (2023); Cheng et al. (2022) loss $\mathcal{L}_{seg}$ for both PV and BEV branches, aiding in extracting discriminative lane features in each view; 2) Depth estimation for the PV branch, which guides effective 3D structure-aware feature extraction of $\mathbf{F}_{pv}$. Depth labels are generated from Li-DAR points, and the loss $\mathcal{L}_{depth}$ is calculated following BEVDepth Li et al. (2022c).

**3D Lane Prediction and Loss.** As we adopt point-based queries $\mathbf{Q} \in \mathbb{R}^{(N \times M) \times C}$, each query naturally corresponds to a 3D point, and every group of $M$ points constructs a complete 3D lane. Thus, we predict x, z, and visibility for each point query on the predefined y coordinates Chen et al. (2022); Luo et al. (2023) and a classification probability for each lane. Overall, the total loss is:

$$\mathcal{L}_{lane} = w_x \mathcal{L}_x + w_z \mathcal{L}_z + w_v \mathcal{L}_v + w_c \mathcal{L}_c,$$
$$\mathcal{L}_{aux} = w_{seg} \mathcal{L}_{seg} + w_{depth} \mathcal{L}_{depth},$$
$$\mathcal{L}_{total} = \mathcal{L}_{lane} + \mathcal{L}_{aux}.$$

where $w_*$ denotes different loss weights. We adopt the L1 loss $\mathcal{L}_x$ and $\mathcal{L}_z$ to learn the x, z positions, focal loss Lin et al. (2017) $\mathcal{L}_c$ to learn the lane category, and BCELoss $\mathcal{L}_v$ to learn visibility.

# 4 EXPERIMENTS

## 4.1 DATASETS

We evaluate our method on OpenLane Chen et al. (2022), the *sole* public 3D lane dataset featuring multi-modal sources, OpenLane is a large-scale dataset built on Waymo Open Dataset Sun et al. (2020), comprising 200K frames and 880K lanes across six driving scenarios and 14 lane categories. The LiDAR data, collected using 64-beam LiDARs, is sampled at 10Hz. This extensive dataset provides a solid foundation for evaluating 3D lane algorithms comprehensively.

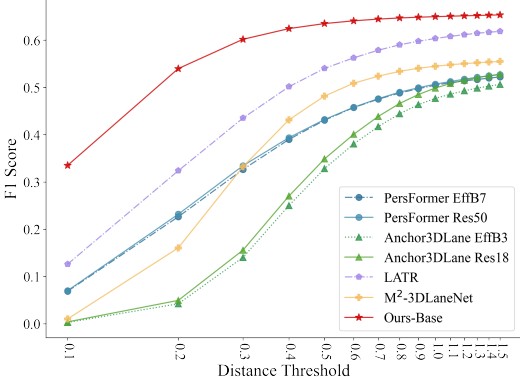

Figure 6: **F1 score vs. Distance Threshold.** Our method consistently achieves superior results under more stringent criteria.

## 4.2 METRICS

We adopt the evaluation metrics established by OpenLane Chen et al. (2022), framing 3D lane detection evaluation as a matching problem based on the edit distance between predictions and ground truth. Successful matching results in computed metrics, including F-Score, category accuracy, and error in X/Z-axes. A successful match for each predicted 3D lane is defined when at least 75% of its points have a distance to the ground truth below the predefined threshold $D_{thre}$.

## 4.3 IMPLEMENTATION DETAILS

**Models.** In the base version of DV-3DLane, we employ ResNet34 He et al. (2016) and Pillar-Net34 Shi et al. (2022) as the backbones for our camera and LiDAR branches, respectively. For the lite version, we utilize ResNet18 and PillarNet18. The base version features two decoder layers, while the lite version employs a single decoder layer. Following LATR Luo et al. (2023), we set the number of lane queries to 40, and we employ deformable attention with 4 heads, 8 sample points, and 256 embedding dimensions.

**Training.** We use the Adam optimizer Kingma & Ba (2014) with a weight decay of 0.01. The learning rate is set to 2e-4, and our models undergo training for 24 epochs with a batch size of 32. We employ the cosine annealing scheduler Loshchilov & Hutter (2016) with $T_{max} = 8$. Our input images are of resolution 720×960, and we adopt a voxel size of (0.2m, 0.4m) for the X and Y axes.

## 4.4 MAIN RESULTS

It's important to note that the existing metrics use a rather *lenient* distance threshold of $D_{thre}$=**1.5m**. However, in the context of ensuring safety in AD, this value, although commonly used for assessment purposes, may be considered overly permissive. Following M²-3DLaneNet Luo et al. (2022), we extend our evaluation to include a *more stringent* threshold, $D_{thre}$=**0.5m**. Further, we illustrate the relationship between the F1 score performance and different distance thresholds for various models, as shown in Figure 6. Notably, our method consistently achieves superior results, even when evaluated under a *much more stringent* criterion of $D_{thre}$=**0.1m**. In contrast, other approaches experience a noticeable decline in performance as the distance threshold decreases. These findings confirm the robustness of our method across varying distance thresholds, particularly highlighting its advantage in precise localization.

We present the main results in Table 1, obtained from experiments conducted on the OpenLane-1K dataset. The evaluation uses both $D_{thre}$=1.5m and $D_{thre}$=0.5m criteria, allowing for a comprehensive and insightful comparison. It is evident that DV-3DLane consistently outperforms previous state-of-the-art (SoTA) methods across all metrics. Notably, when applying a more strict 0.5m threshold, DV-3DLane demonstrates a substantial **11.2%** improvement in the F1 score. It is noteworthy that our method excels in localization accuracy, leading to significant performance improvements. Specifically, our method achieves remarkable reductions in localization errors: 52%/50% for

Table 1: Comprehensive 3D Lane evaluation comparison on OpenLane with variable metrics. † denotes the results obtained using their provided models. "Image-Branch" and "LiDAR-Branch" refer to our image and LiDAR branches, respectively. "LATR + LiDAR" denotes the model that combines the SOTA method LATR with LiDAR input, projecting all points into the image space and using them as additional features in the network.

| Dist. | Methods | Backbone | Modality | F1 ↑ | Acc. ↑ | X error (m) ↓ | | Z error (m) ↓ | |
|---|---|---|---|---|---|---|---|---|---|
| | | | | | | near | far | near | far |
| 1.5 m | 3DLaneNet Garnett et al. (2019) | VGG-16 | C | 44.1 | - | 0.593 | 0.494 | 0.140 | 0.195 |
| | GenLaneNet Guo et al. (2020) | ERFNet | C | 32.3 | - | 0.591 | 0.684 | 0.411 | 0.521 |
| | PersFormer Chen et al. (2022) | EffNet-B7 | C | 50.5 | 89.5 | 0.319 | 0.325 | 0.112 | 0.141 |
| | Anchor3DLane Huang et al. (2023)† | EffNet-B3 | C | 52.8 | 89.6 | 0.408 | 0.349 | 0.186 | 0.143 |
| | M²-3DLaneNet Luo et al. (2022) | EffNet-B7 | C+L | 55.5 | 88.2 | 0.283 | 0.256 | 0.078 | 0.106 |
| | Anchor3DLane Huang et al. (2023)† | ResNet-18 | C | 50.7 | 89.3 | 0.422 | 0.349 | 0.188 | 0.146 |
| | PersFormer Chen et al. (2022) | ResNet-50 | C | 52.7 | 88.4 | 0.307 | 0.319 | 0.083 | 0.117 |
| | LATR Luo et al. (2023) | ResNet-50 | C | 61.9 | 92.0 | 0.219 | 0.259 | 0.075 | 0.104 |
| | DV-3DLane-Tiny (Ours) | ResNet-18 | C+L | 63.4 | 91.6 | 0.137 | 0.159 | 0.034 | 0.063 |
| | DV-3DLane-Base (Ours) | ResNet-34 | C+L | 65.4 | 92.4 | 0.118 | 0.131 | 0.032 | 0.053 |
| | DV-3DLane-Large (Ours) | ResNet-50 | C+L | **66.8** | **93.3** | **0.115** | **0.134** | **0.029** | **0.049** |
| | *Improvement* | - | - | ↑4.9 | ↑1.3 | ↓0.104 | ↓0.122 | ↓0.046 | ↓0.055 |
| 0.5 m | PersFormer Chen et al. (2022) | EffNet-B7 | C | 36.5 | 87.8 | 0.343 | 0.263 | 0.161 | 0.115 |
| | Anchor3DLane Huang et al. (2023)† | EffNet-B3 | C | 34.9 | 88.5 | 0.344 | 0.264 | 0.181 | 0.134 |
| | M²-3DLaneNet Luo et al. (2022) | EffNet-B7 | C+L | 48.2 | 88.1 | 0.217 | 0.203 | 0.076 | 0.103 |
| | Anchor3DLane Huang et al. (2023)† | ResNet-18 | C | 32.8 | 87.9 | 0.350 | 0.266 | 0.183 | 0.137 |
| | PersFormer Chen et al. (2022) | ResNet-50 | C | 43.2 | 87.8 | 0.229 | 0.245 | 0.078 | 0.106 |
| | LATR Luo et al. (2023) | ResNet-50 | C | 54.0 | 91.7 | 0.171 | 0.201 | 0.072 | 0.099 |
| | LATR + LiDAR | ResNet-50 | C+L | 57.4 | 92.1 | 0.167 | 0.185 | 0.071 | 0.088 |
| | Image-Branch (Ours) | ResNet-34 | C | 52.9 | 90.3 | 0.173 | 0.212 | 0.069 | 0.098 |
| | LiDAR-Branch (Ours) | PillarN-34 | L | 54.1 | 84.4 | 0.282 | 0.191 | 0.096 | 0.124 |
| | DV-3DLane-Tiny (Ours) | ResNet-18 | C+L | 60.9 | 91.8 | 0.097 | 0.124 | 0.033 | 0.062 |
| | DV-3DLane-Base (Ours) | ResNet-34 | C+L | 63.5 | 92.4 | 0.090 | 0.102 | 0.031 | 0.053 |
| | DV-3DLane-Large (Ours) | ResNet-50 | C+L | **65.2** | **93.4** | **0.082** | **0.101** | **0.028** | **0.048** |
| | *Improvement* | - | - | ↑11.2 | ↑1.7 | ↓0.089 | ↓0.100 | ↓0.044 | ↓0.051 |

X near/far, and 61%/52% for Z near/far. Due to space limitations, results in various scenarios and studies about robustness concerning calibration noise are included in our Appendix.

**Effect of Multiple Modalities.** To explore the impact of individual modalities, we conduct experiments using single modalities, as outlined in the "Image-Branch" and "LiDAR-Branch" rows of Table 1. The results illustrate that DV-3DLane significantly enhances performance compared to using images alone or relying solely on LiDAR data. Notably, our method significantly surpasses configurations that simply equip LATR with LiDAR input across all metrics, underscoring the substantial improvements achieved by DV-3DLane in leveraging information from both modalities. Moreover, to evaluate the effect of **dual-view**, we conduct experiments using single-modality input but transforming features extracted from the backbone into another view, yielding single-modal dual-view features. Then, our dual-view decoder is applied, and the results are detailed in our Appendix. Additionally, we conduct experiments using our "Image-Branch" on the Apollo Guo et al. (2020) dataset, which exclusively contains image data. The results are provided in our Appendix.

**Qualitative Results.** We present a qualitative comparison between DV-3DLane and LATR Luo et al. (2023) in Figure 7, demonstrating that our method achieves more robust and accurate predictions across various scenarios. More visualization results are included in our Appendix.

## 4.5  ABLATION STUDIES

We conduct all ablation studies on OpenLane-300 following established practices Chen et al. (2022); Luo et al. (2023); Huang et al. (2023), while adopting a **0.5m** threshold $D_{thre}$ for evaluation.

**Effect of Bidirectional Feature Fusion.** The corresponding experiments are included in our Appendix, due to space limitations. We kindly direct the readers to refer to the Appendix for details. The results confirm the effectiveness of the proposed bidirectional feature fusion approach.

**Effect of Unified Query.** We study the effect of our unified queries generation strategy in Table 2, where "Random" means random initialization using `nn.Embedding`, "$\mathbf{Q}_{pv}$" denotes using only PV queries, and "$\mathbf{Q}_{bev}$" refers to using only BEV queries. Replacing our unified queries with randomly initialized ones Carion et al. (2020); Zhu et al. (2020); Li et al. (2022d) results in a decrease of 1.0 in the F1 score compared to our approach. Interestingly, employing a single space instance-

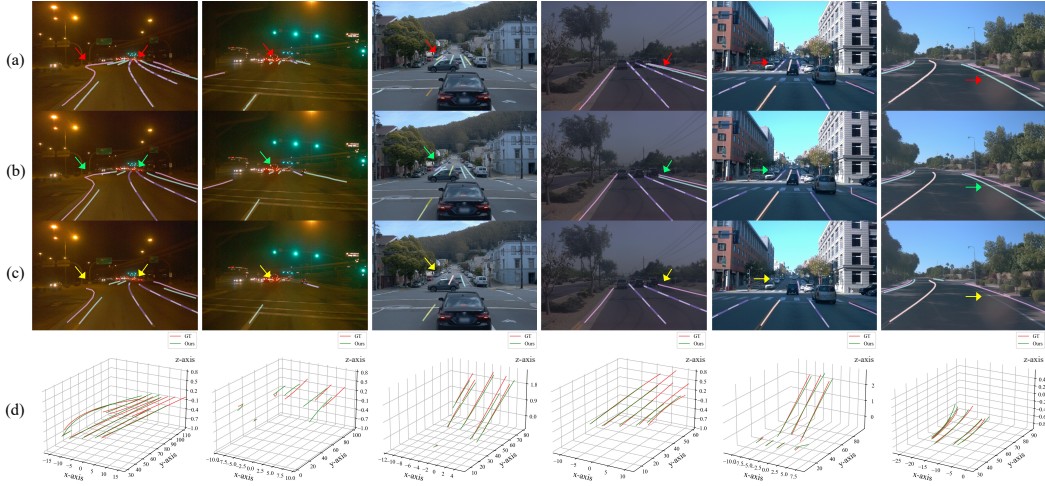

Figure 7: **Qualitative Results.** We present the projection of 3D lanes from ground truth, predictions of DV-3DLane and the SOTA method LART Luo et al. (2023) in rows (a), (b), (c), respectively. Row (d) depicts the comparison between ground truth (red) and ours (green) in 3D space. We highlight the differences with colored arrows. Best viewed in color and zoom in for details.

aware query yields even lower F1 scores of 69.6%/69.1% for PV/BEV, respectively, than random initialization. This underscores the inadequacy of a single-space lane-aware query in capturing complex 3D lane features comprehensively existing in both PV and BEV spaces. However, our dual-view strategy, generating lane-aware queries *w.r.t.* both views, improves overall performance to 70.7, achieving the best result. This demonstrates that our method effectively integrates the strengths of features from two spaces, forming a cohesive query set.

**Effect of 3D Dual-view Deformable Attention.** To evaluate the efficacy of our proposed Dual-view Deformable Attention, we conduct ablation studies in Table 3, where "PV space" and "BEV space" mean using single space in the decoder. "DeepInteration" Yang et al. (2022) denotes sequential fusion of features from different spaces, and "FUTR3D" Chen et al. (2023) refer to a modality-agnostic approach where sampling locations differ across views. We compare DV-3DLane against alternative approaches, including single-view fused method, as well as methods proposed in Deep-Interation and FUTR3D, as described in Section 3.3. The results underscore the significance of our approach. In detail, sampling only PV space features leads to a notable drop (70.7→63.6) in performance, showing the importance of BEV space due to its advantages in localization. Besides, our method outperforms the sequential approach of DeepInteration with a substantial 2.0 gain in F1 score. Furthermore, compared to the modality-agnostic approach proposed in FUTR3D, our method achieves a 0.5 improvement, emphasizing the importance of consistent sampling locations in deformable attention across different spaces.

Table 2: Effect of unified query.

| Methods | F1 | X error (m) | | Z error (m) | |
|---|---|---|---|---|---|
| | | *near* | *far* | *near* | *far* |
| Random | 69.7 | 0.123 | 0.151 | 0.059 | 0.081 |
| $\mathbf{Q}_{pv}$ | 69.6 | 0.124 | 0.155 | 0.059 | 0.079 |
| $\mathbf{Q}_{bev}$ | 69.1 | 0.122 | 0.145 | 0.058 | 0.077 |
| Ours | 70.7 | 0.123 | 0.146 | 0.058 | 0.078 |

Table 3: Effect of 3D dual-view deformable attention.

| Methods | F1 | X error (m) | | Z error (m) | |
|---|---|---|---|---|---|
| | | *near* | *far* | *near* | *far* |
| PV space | 63.6 | 0.150 | 0.202 | 0.060 | 0.081 |
| BEV space | 68.5 | 0.127 | 0.151 | 0.064 | 0.087 |
| DeepInteration | 68.7 | 0.126 | 0.157 | 0.059 | 0.081 |
| FUTR3D | 70.2 | 0.118 | 0.145 | 0.057 | 0.077 |
| Ours | 70.7 | 0.123 | 0.146 | 0.058 | 0.078 |

## 5 CONCLUSION

In this work, we introduce DV-3DLane, a novel end-to-end multi-modal 3D lane detection framework that leverages the strengths of both PV and BEV spaces. To this end, we propose three novel modules that effectively utilize dual-view representation on different levels, consistently enhancing performance. Extensive experiments substantiate the outstanding advancements achieved by DV-3DLane, establishing a new state of the art on OpenLane.

## ACKNOWLEDGMENTS

This work was supported by NSFC with Grant No. 62293482, by the Basic Research Project No. HZQB-KCZYZ-2021067 of Hetao Shenzhen HK S&T Cooperation Zone, by Shenzhen General Program No. JCYJ20220530143600001, by Shenzhen-Hong Kong Joint Funding No. SGDX20211123112401002, by the National Key R&D Program of China with grant No. 2018YFB1800800, by the Shenzhen Outstanding Talents Training Fund 202002, by Guangdong Research Project No. 2017ZT07X152 and No. 2019CX01X104, by the Guangdong Provincial Key Laboratory of Future Networks of Intelligence (Grant No. 2022B1212010001), by the Guangdong Provincial Key Laboratory of Big Data Computing, The Chinese University of Hong Kong, Shenzhen, by the NSFC 61931024&12326610, by the Shenzhen Key Laboratory of Big Data and Artificial Intelligence (Grant No. ZDSYS201707251409055), and the Key Area R&D Program of Guangdong Province with grant No. 2018B03033800, by Tencent&Huawei Open Fund.

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
