# A  APPENDIX

## A.1  MODEL COMPLEXITY

As stated in our main paper, DV-3DLane achieves SoTA performance, and its lite version also surpasses all previous methods in terms of F1 score and localization errors, while achieving an impressive FPS of 13.49. In this section, we study the model complexity, as shown in Table 1. Our base model achieves a competitive FPS of 8.82 while maintaining a strong F1 score of 63.5. Notably, our tiny version excels with an FPS of 13.49, along with a notable F1 score of 60.9.

Table 1: **Model complexity.** FPS is evaluated on a single V100 GPU.

| Model | Backbone | FPS | F1 |
|---|---|---|---|
| PersFormer | Efficient-B7 | 11.67 | 36.5 |
| PersFormer | Res50 | 9.48 | 43.2 |
| M$^2$-3DLaneNet | Efficient-B7 | 6.48 | 48.2 |
| Anchor3DLane | Efficient-B3 | 3.07 | 34.9 |
| Anchor3DLane | Res18 | 3.45 | 32.8 |
| LATR | Res50 | 13.34 | 54.0 |
| DV-3DLane-Tiny | Res18, PillarNet18 | 13.49 | 60.9 |
| DV-3DLane-Base | Res34, PillarNet34 | 8.82 | 63.5 |
| DV-3DLane-Large | Res50, PillarNet34 | 6.18 | 65.2 |

## A.2  SCENARIO STUDIES

Additionally, we comprehensively evaluated DV-3DLane across *diverse scenarios* within OpenLane. As depicted in Table 2, our method consistently outperforms all previous approaches across all six challenging scenarios by a large margin. Visualizations are provided in Figure 1. Overall, these results reveal the effectiveness of our design.

Table 2: Comparison with other 3D lane detection methods on the OpenLane validation dataset. † denotes that the results are obtained using their provided models.

| Dist. | Methods | Backbone | Modality | All | Up & Down | Curve | Extreme Weather | Night | Intersection | Merge & Split |
|---|---|---|---|---|---|---|---|---|---|---|
| 1.5 m | 3DLaneNet Garnett et al. (2019) | VGG-16 | C | 44.1 | 40.8 | 46.5 | 47.5 | 41.5 | 32.1 | 41.7 |
| | GenLaneNet Guo et al. (2020) | ERFNet | C | 32.3 | 25.4 | 33.5 | 28.1 | 18.7 | 21.4 | 31.0 |
| | PersFormer Chen et al. (2022) | EffNet-B7 | C | 50.5 | 42.4 | 55.6 | 48.6 | 46.6 | 40.0 | 50.7 |
| | Anchor3DLane Huang et al. (2023)† | EffNet-B3 | C | 52.8 | 48.5 | 50.7 | 56.9 | 43.6 | 48.5 | 50.7 |
| | M$^2$-3DLaneNet Luo et al. (2022) | EffNet-B7 | C+L | 55.5 | 53.4 | 60.7 | 56.2 | 51.6 | 43.8 | 51.4 |
| | PersFormer Chen et al. (2022) | ResNet-50 | C | 52.7 | 46.4 | 57.9 | 52.9 | 47.2 | 41.6 | 51.4 |
| | LATR Luo et al. (2023) | ResNet-50 | C | 61.9 | 55.2 | 68.2 | 57.1 | 55.4 | 52.3 | 61.5 |
| | Anchor3DLane Huang et al. (2023)† | ResNet-18 | C | 50.7 | 45.3 | 53.7 | 48.5 | 51.6 | 45.3 | 48.5 |
| | DV-3DLane-Tiny | ResNet-18 | C+L | 63.4 | 59.9 | 69.8 | 62.2 | 58.8 | 53.5 | 60.6 |
| | DV-3DLane-Base | ResNet-34 | C+L | 65.4 | 60.9 | **72.1** | 64.5 | 61.3 | 55.5 | 61.6 |
| | DV-3DLane-Large | ResNet-50 | C+L | **66.8** | **61.1** | 71.5 | **64.9** | **63.2** | **58.6** | **62.8** |
| | *Improvement* | - | - | ↑*4.9* | ↑*5.9* | ↑*3.9* | ↑*7.8* | ↑*7.8* | ↑*6.3* | ↑*1.3* |
| 0.5 m | PersFormer Chen et al. (2022) | EffNet-B7 | C | 36.5 | 26.8 | 36.9 | 33.9 | 34.0 | 28.5 | 37.4 |
| | Anchor3DLane Huang et al. (2023)† | EffNet-B3 | C | 34.9 | 28.3 | 31.8 | 30.7 | 32.2 | 29.9 | 33.9 |
| | M$^2$-3DLaneNet Luo et al. (2022) | EffNet-B7 | C+L | 48.2 | 40.7 | 48.2 | 49.8 | 46.2 | 38.7 | 44.2 |
| | PersFormer Chen et al. (2022) | ResNet-50 | C | 43.2 | 36.3 | 42.4 | 45.4 | 39.3 | 32.9 | 41.7 |
| | LATR Luo et al. (2023) | ResNet-50 | C | 54.0 | 44.9 | 56.2 | 47.6 | 46.2 | 45.5 | 55.6 |
| | Anchor3DLane Huang et al. (2023)† | ResNet-18 | C | 32.8 | 26.5 | 27.6 | 31.2 | 30.0 | 28.1 | 31.7 |
| | DV-3DLane-Tiny | ResNet-18 | C+L | 60.9 | 56.9 | 65.9 | 60.0 | 56.8 | 50.7 | 57.6 |
| | DV-3DLane-Base | ResNet-34 | C+L | 63.5 | 58.6 | **69.3** | 62.4 | 59.9 | 53.9 | 59.3 |
| | DV-3DLane-Large | ResNet-50 | C+L | **65.2** | **59.1** | 69.2 | **63.0** | **62.0** | **56.9** | **60.5** |
| | *Improvement* | - | - | ↑*11.2* | ↑*14.2* | ↑*13.1* | ↑*13.2* | ↑*15.8* | ↑*11.4* | ↑*4.9* |

## A.3  ROBUSTNESS

To investigate the robustness of our model amid calibration noise, given that perfect calibration is not always viable in real-world settings, we conduct experiments incorporating diverse levels of calibration noise to understand the model's performance under noisy conditions.

**Noise settings:**   Following the methodology of Yu et al. (2023), we introduce two noise settings: **Noise (N)** and **Stronger Noise (SN)**. In 'Noise (N)', we introduce random rotations within [1°, 5°] and translations within [0.5cm, 1.0cm] to the calibration. For 'Stronger Noise (SN)', these parameters are doubled to reflect stronger calibration disturbances.

**Results without training noise:**   We first test our model, which has *not* been trained with additional noise, under these noisy conditions. The results, presented in the first row of Table 3, show a

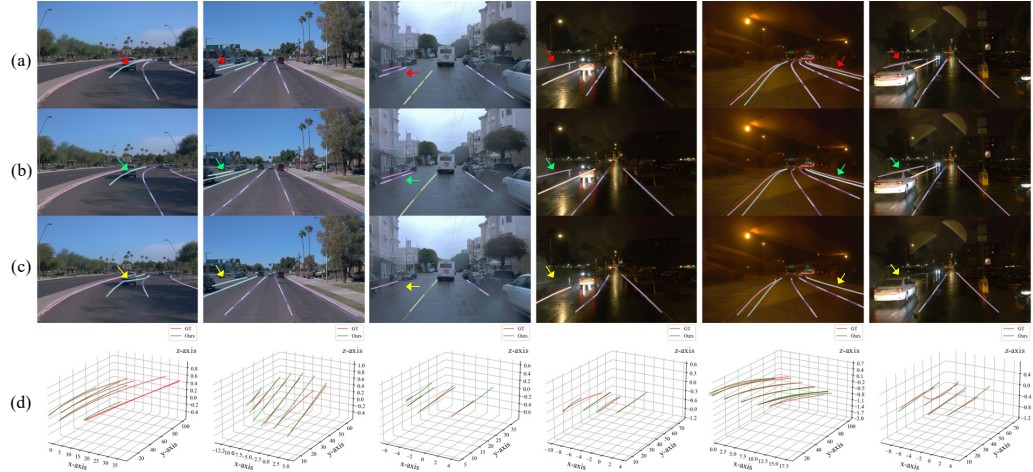

Figure 1: **More Results.** Rows (a), (b), (c) show projections of 3D lanes from the ground truth (GT), DV-3DLane, and LATR Luo et al. (2023), with differences highlighted by colored arrows. Row (d) compares GT (red) and our prediction (green) in 3D. Best viewed in color and zoom in for details.

notable decline in performance as the intensity of noise increases. Specifically, under a probability setting of 0.7, the performance deteriorates from 63.5 to 32.4/31.4 in the 'Noise'/'Stronger Noise' settings.

**Enhancing robustness via training:** To enhance robustness, we incorporate calibration noise during the training phase. This strategy substantially mitigates the performance degradation caused by noisy calibration, as shown in the second and third rows of Table 3.

**Comparative analysis:** In comparison to the baseline (first row), we can observe that training with calibration noise significantly strengthens the robustness of our model. It effectively maintains comparable results under noisy calibration conditions. Additionally, the model trained with 'Stronger Noise' exhibits greater robustness compared to the one trained with less intense noise, underscoring the benefits of this training strategy.

Table 3: Impact of noise on calibration parameters. We set two noise levels in the experiments, "**Noise (N)**" and "**Stronger Noise (SN)**". In the **Train** column, "-" denotes *no* noise is added during the training phase. "Prob" denotes the probability of adding the corresponding noise into the training/eval phases. Each result group consists of `F1-score / Accuracy`.

| Train | | Eval | | | | | | |
|---|---|---|---|---|---|---|---|---|
| @noise (N/SN) | | ———— | + Noise (N) | | | + Stronger Noise (SN) | | |
| | | Prob=0.0 | Prob=0.3 | Prob=0.5 | Prob=0.7 | Prob=0.3 | Prob=0.5 | Prob=0.7 |
| - | Prob=0.0 | 63.5 / 92.4 | 52.2 / 89.9 | 40.9 / 85.6 | 32.4 / 82.5 | 52.0 / 89.0 | 40.3 / 83.4 | 31.4 / 79.2 |
| **N** | Prob=0.3 | 63.0 / 93.1 | 62.5 / 92.9 | 62.0 / 92.9 | 61.5 / 92.9 | 62.2 / 92.9 | 61.5 / 92.9 | 60.8 / 92.7 |
| **SN** | Prob=0.3 | 63.4 / 92.5 | 62.8 / 92.4 | 62.3 / 92.3 | 61.8 / 92.2 | 62.7 / 92.4 | 62.1 / 92.2 | 61.7 / 92.2 |

## A.4 EFFECT OF DUAL-VIEW

Apart from studying the impact of multiple modalities, we conducted experiments on OpenLane-1K dataset to analyze the effect of the dual views, providing a comprehensive understanding of our approach. As shown in Table 4, we conducted two sets of experiments: **1)** Using single modality and single view. **2)** Using single modality but dual views.

In the first set of experiments, rows #1 and #2 present the performance using individual modalities.

In the second set of experiments:

- For the image branch experiment, we adopt a strategy similar to BEVFormer Li et al. (2022b), utilizing deformable attention to transform image features into BEV features. Then, we apply our dual-view decoder upon this, and the outcomes are illustrated in row #3 of Table 4.
- For the LiDAR branch experiment, we project LiDAR point cloud features onto the 2D image plane to generate perspective-view features. The results of this approach are presented in row #4 of Table 4.

The results in Table 4 underscore that the dual-view representation significantly enhances the performance of baseline models in single-modal scenarios (comparing #1 with #3 and #2 with #4). This improvement confirms the effectiveness of our dual-view approach in learning 3D lane detection. Most notably, the combination of image and LiDAR modalities, coupled with our dual-view representation, achieves the best results, as shown in row #5. This synergy of modalities underlines the superiority of our proposed method.

Table 4: Comparison of single and dual-view approaches on OpenLane-1K dataset with 0.5m setting.

| # Line | Inputs | View | Backbone | F1 | Acc. | X error (m) near \| far | Z error (m) near \| far |
|--------|--------|------|----------|-----|------|---------------------------|---------------------------|
| #1 | Image | PV | Res34 | 52.9 | 90.3 | 0.173 \| 0.212 | 0.069 \| 0.098 |
| #2 | LiDAR | BEV | PillarNet34 | 54.1 | 84.4 | 0.282 \| 0.191 | 0.096 \| 0.123 |
| #3 | Image | Dual Views | Res34 | 54.3 | 91.5 | 0.165 \| 0.200 | 0.067 \| 0.094 |
| #4 | LiDAR | Dual Views | PillarNet34 | 55.3 | 87.9 | 0.156 \| 0.143 | 0.031 \| 0.050 |
| #5 | DV-3DLane | Dual Views | Res34+PillarNet34 | 63.5 | 92.4 | 0.090 \| 0.102 | 0.031 \| 0.053 |

## A.5 EFFECT OF BIDIRECTIONAL FEATURE FUSION.

To validate the effectiveness of this strategy, we compare the performance of our method with the other three fusion design choices, as shown in Table 5, where "Cam" means only image features in PV branch, and "LiDAR" denotes only point features in BEV branch. "L→C" denotes the LiDAR to camera fusion for PV branch, and conversely, "C→L" denotes the camera to LiDAR fusion for BEV branch. It shows that the absence of fusion leads to the poorest performance (#1). Further, employing one-way fusion, either from camera to LiDAR (#2) or LiDAR to camera (#3), results in 1.2% and 1.1% improvements, respectively *w.r.t.* non-fusion (#1). Remarkably, our bidirectional fusion (#4) yields the highest performance, a 2.8% gain in F1. This improvement highlights the efficacy of our strategy in effectively leveraging multi-modal features in both PV and BEV spaces.

Table 5: Effect of bidirectional feature fusion.

| # Line | Methods | F1 | X error (m) near \| far | Z error (m) near \| far |
|--------|---------|-----|---------------------------|---------------------------|
| #1 | Cam & LiDAR | 67.9 | 0.133 \| 0.157 | 0.060 \| 0.083 |
| #2 | Cam & C→L | 69.1 | 0.135 \| 0.151 | 0.060 \| 0.081 |
| #3 | L→C & LiDAR | 69.0 | 0.130 \| 0.156 | 0.059 \| 0.078 |
| #4 | L→C & C→L | 70.7 | 0.123 \| 0.146 | 0.058 \| 0.078 |

## A.6 IMAGE BRANCH ON APOLLO

Table. 6 illustrates the results of our image branch on the Apollo dataset Guo et al. (2020), compared with existing methods.

Table 6: **Results on Apollo 3D Synthetic dataset.** "Image-Branch" denotes the image branch of our DV-3DLane.

| Scene | Methods | F1 ↑ | AP ↑ | X error (m) ↓ | | Z error (m) ↓ | |
|---|---|---|---|---|---|---|---|
| | | | | *near* | *far* | *near* | *far* |
| Balanced Scene | 3DLaneNet Garnett et al. (2019) | 86.4 | 89.3 | 0.068 | 0.477 | 0.015 | 0.202 |
| | Gen-LaneNet Guo et al. (2020) | 88.1 | 90.1 | 0.061 | 0.496 | 0.012 | 0.214 |
| | CLGo Liu et al. (2022) | 91.9 | 94.2 | 0.061 | 0.361 | 0.029 | 0.250 |
| | PersFormer Chen et al. (2022) | 92.9 | - | 0.054 | 0.356 | 0.010 | 0.234 |
| | GP Li et al. (2022a) | 91.9 | 93.8 | 0.049 | 0.387 | 0.008 | 0.213 |
| | CurveFormer Bai et al. (2022) | 95.8 | 97.3 | 0.078 | 0.326 | 0.018 | 0.219 |
| | Anchor3DLane Huang et al. (2023) | 95.6 | 97.2 | 0.052 | 0.306 | 0.015 | 0.223 |
| | LATR Luo et al. (2023) | 96.8 | 97.9 | 0.022 | 0.253 | 0.007 | 0.202 |
| | Image-Branch (Ours) | 96.4 | 97.6 | 0.046 | 0.299 | 0.016 | 0.213 |
| Rare Subset | 3DLaneNet Garnett et al. (2019) | 74.6 | 72.0 | 0.166 | 0.855 | 0.039 | 0.521 |
| | Gen-LaneNet Guo et al. (2020) | 78.0 | 79.0 | 0.139 | 0.903 | 0.030 | 0.539 |
| | CLGo Liu et al. (2022) | 86.1 | 88.3 | 0.147 | 0.735 | 0.071 | 0.609 |
| | PersFormer Chen et al. (2022) | 87.5 | - | 0.107 | 0.782 | 0.024 | 0.602 |
| | GP Li et al. (2022a) | 83.7 | 85.2 | 0.126 | 0.903 | 0.023 | 0.625 |
| | CurveFormer Bai et al. (2022) | 95.6 | 97.1 | 0.182 | 0.737 | 0.039 | 0.561 |
| | Anchor3DLane Huang et al. (2023) | 94.4 | 96.9 | 0.094 | 0.693 | 0.027 | 0.579 |
| | LATR Luo et al. (2023) | 96.1 | 97.3 | 0.050 | 0.600 | 0.015 | 0.532 |
| | Image-Branch (Ours) | 95.6 | 97.2 | 0.071 | 0.664 | 0.025 | 0.568 |
| Visual Variations | 3DLaneNet Garnett et al. (2019) | 74.9 | 72.5 | 0.115 | 0.601 | 0.032 | 0.230 |
| | Gen-LaneNet Guo et al. (2020) | 85.3 | 87.2 | 0.074 | 0.538 | 0.015 | 0.232 |
| | CLGo Liu et al. (2022) | 87.3 | 89.2 | 0.084 | 0.464 | 0.045 | 0.312 |
| | PersFormer Chen et al. (2022) | 89.6 | - | 0.074 | 0.430 | 0.015 | 0.266 |
| | GP Li et al. (2022a) | 89.9 | 92.1 | 0.060 | 0.446 | 0.011 | 0.235 |
| | CurveFormer Bai et al. (2022) | 90.8 | 93.0 | 0.125 | 0.410 | 0.028 | 0.254 |
| | Anchor3DLane Huang et al. (2023) | 91.4 | 93.6 | 0.068 | 0.367 | 0.020 | 0.232 |
| | LATR Luo et al. (2023) | 95.1 | 96.6 | 0.045 | 0.315 | 0.016 | 0.228 |
| | Image-Branch (Ours) | 91.3 | 93.4 | 0.095 | 0.417 | 0.040 | 0.320 |