# OpenReview forum: "DV-3DLane: End-to-end Multi-modal 3D Lane Detection with Dual-view Representation"
_ICLR.cc/2024/Conference — ICLR 2024 poster_

### Official Review · Reviewer_9oXi · 2023-10-27

**Soundness:** 4 excellent
**Presentation:** 4 excellent
**Contribution:** 4 excellent
**Rating:** 10
**Confidence:** 5

**Summary:**

This paper covers lane detection task using attention mechanism from multi-modal queries of vision and LiDAR.

**Strengths:**

* Really novel idea that does this task of lane detection using query clustering from multiple perspective.
* SOTA results, and code should be available online soon
* Takes care of runtime optimization as well.

**Weaknesses:**

NA

**Questions:**

NA

---

> ### Author Response · Authors · 2023-11-20
> **Response to Reviewer 9oXi**
>
> Thank you for your encouraging comments and for recognizing the strengths of our work.
> We appreciate your acknowledgment of the "Really novel idea that does this task of lane detection using query clustering from multiple perspective", "SOTA results" and your observation regarding "runtime optimization". We assure you that we will release our code.

---

### Official Review · Reviewer_Z8i9 · 2023-10-31

**Soundness:** 3 good
**Presentation:** 2 fair
**Contribution:** 1 poor
**Rating:** 3
**Confidence:** 5

**Summary:**

This paper presents a multi-modal 3D lane detection method. A bidirectional feature fusion approach is proposed to incorporate multi-modal features into each view space. A unified query generation module is adopted that provides lane-aware prior knowledge from both views. A 3D dual-view deformable attention mechanism that combines discriminative features from both PV and BEV into queries for precise 3D lane detection. Comprehensive experiments on the public benchmark, OpenLane, prove the efficacy and efficiency of DV-3DLane.

**Strengths:**

*  The performance of this article exceeded that of existing methods. A significant increase was also achieved in the multimodal feature fusion at the backbone level.
*  The paper is well written and easy to follow.

**Weaknesses:**

*  Most methods in Table 1 are based on camera. As can be seen from the results in Table 3, the performance of camera-only is not competitive. Multimodal inputs should be introduced to some methods for a comparative performance evaluation. Otherwise, presenting multimodal fusion as a key contribution is somewhat insufficient.

* In Table 4, the performance of the queries generated based on PV and BEV is not as high as random queries, indicating that the adaptive generation of queries doesn't work. Although the queries after clustering have achieved performance improvement, it remains to be seen whether this improvement is brought about by the extra network.

* The feature fusion in the backbone and decoder is quite tricky, making it difficult to be viewed as a major contribution point.

**Questions:**

*  The main performance improvement primarily comes from multi-modalities, but multi-modalities inherently achieve higher points than a single modality. Therefore, it would be best to prove that the multi-modal performance of existing methods is inferior to this paper. The camera result is not much competitiveness, especially when compared to some recent methods, such as Group Lane.
*  Using Deformable attention to aggregate multi-modal features lacks innovation.

---

> ### Author Response · Authors · 2023-11-20
> **Response to Reviewer Z8i9 [1/3]**
>
> We thank the reviewer for throwing time into reviewing our paper. To address the reviewer's concerns and questions, we would like to first highlight our contributions:
>
> **1)** To bridge the gap of multi-modal exploration in 3D lane detection, we propose the DV-3DLane, an end-to-end multi-modal 3D lane detection framework. To the best of our knowledge, this is **the first end-to-end multi-modal 3D lane detection** framework.
> Besides exploring multiple modalities, our method also harnesses the power of both Perspective View (PV) and Bird's Eye View (BEV) representations. All four other reviewers acknowledge this contribution.
>
> **2)** To achieve the goal of end-to-end multi-model 3D lane detection: First, we propose to fuse multi-modal features bidirectionally among PV and BEV spaces and further generate a lane-aware query set within each space. Then, to unify these two query sets, from two spaces, we propose a **lane-centric clustering** strategy. Afterward, to effectively aggregate PV features and BEV features into the unified queries, we propose the 3D DV Deformable Attention to consistently sample features from two different spaces via 3D deformed points, which are generated by predicting both 3D reference points and their 3D offsets.
>
> **3)** We conduct comprehensive experiments on the OpenLane benchmark and our method significantly improves the performance, compared with the previous multi-modal approach M$^2$-3DLaneNet and the SOTA LATR. Notably, we significantly **cut the errors** in both X and Z directions within near/far ranges by **about 50%**. Moreover, we propose to use a more strict evaluation metric, 0.5m (previous 1.5m), to study the problem in a nuanced manner, towards enhancing real-world driving safety.
>
> Below, please find our response to each of the comments and questions:
>
> **Presentation: 2 fair**
> This contradicts your 2nd strength point:
>  > The paper is well written and easy to follow.
>
> And there is no specific weak point or question about the presentation. We, therefore, kindly appeal the reviewer’s re-evaluation of the “presentation” rating and final rating.
>
> **W1-1: *Regariding performance results in Table 3.***
>
> First, we highlight that Table.3 does not compare the camera-only method and the multi-modal one, as stated by the reviewer.
> Instead, Table 3 focuses on comparing various designs in the context of multi-modal fusion, with **each design incorporating both LiDAR and camera data**.
> The results in Table.3 highlights the effectiveness of our bidirectional fusion within dual-view spaces compared to non-fusion/unidirectional-fusion ones (during the backbone stage).
>
> As highlighted by the reviewer's observation from Table 1, the majority of methods in the literature lean towards a single-modality solution, with M$^2$-3DLaneNet being the exception, albeit now surpassed by subsequent works like LATR.
> Recognizing the significance of point cloud data, in tasks like 3D detection, we aim to explore the underexplored territory of combining LiDAR and image data for 3D lane detection. Our work addresses this gap in research (as mentioned by reviewer 7Rt7), emphasizing the relevance and importance of leveraging both modalities in this field.
>
> **W1-2: *Introduce LiDAR to some methods for a comparative performance evaluation.***
>
> First, we highlight that modifying other methods into multi-modal ones is **non-trivial**. Besides, we compared M$^2$-3DLaneNet, which compared several multi-modal fusion approaches, *e.g.,* PointPillar, BEVFusion results with inferior performance than M$^2$-3DLaneNet. The reviewer has acknowledged the multi-modal lack of research.
>
> To specifically address your concern, we integrated the SOTA method **LATR with LiDAR** inputs, projecting all points into the image and using them as additional features in the network. The results clearly demonstrate the positive impact of LiDAR, although it falls below the performance achieved by our proposed method. This comparison underscores the unique contribution of our work.
> | Model       | Backbone | F1   | Acc. | X error       | Z error       |
> |-------------|----------|------|------|---------------|---------------|
> | LATR        | Res50    | 54.0 | 91.7 | 0.171 / 0.201 | 0.072 / 0.099 |
> | LATR+LiDAR  | Res50    | 57.4 | 92.1 | 0.167 / 0.185 | 0.071 / 0.088 |
> | DV-3DLane   | Res34    | 63.5 | 92.4 | 0.090 / 0.102 | 0.031 / 0.053 |
> | DV-3DLane   | Res50    | 65.2 | 93.4 | 0.082 / 0.101 | 0.028 / 0.048 |

---

> ### Author Response · Authors · 2023-11-20
> **Response to Reviewer Z8i9 [2/3]**
>
> **W2: *Regarding questions to Table 4.***
> In fact, we have explained this phenomenon in Section 4.5, titled 'Effect of Unified Query,' in the main text. The results indicate that solely using queries from one view does not effectively aggregate features from another view. However, initializing queries with *nn.embedding* allows queries to learn how to match features from different views. Furthermore, by equipping our proposed clustering strategy, the model achieves the best results, which strongly supports the efficacy of our query clustering in integrating multi-modal queries from different view spaces. **We believe this strategy could facilitate other multi-view or/and multi-modal studies as well.**
>
> **W3: *Regarding the contribution of feature fusion in the backbone and decoder.***
>
> We appreciate the reviewer's comment regarding the feature fusion aspect of our work. Our primary goal is **to advance the state of 3D lane detection by introducing an effective end-to-end multi-modal framework**. To achieve this, we have developed the dual-view model to harness feature representations from different views.
>
> While we acknowledge that our work incorporates feature fusion in both the backbone and decoder, it's essential to emphasize that our contribution extends beyond this specific aspect. Although we have demonstrated the effectiveness of our fusion methods compared to alternative approaches, our research is **not solely centered on** the fusion techniques themselves. Instead, it addresses the broader challenge of multi-modal exploration for 3D lane detection and presents innovative solutions to tackle these challenges.
>
> One of our key contributions is the introduction of **lane-centric clustering**, a novel technique designed to cohesively synergize queries from different views. This aspect is integral to our approach and constitutes a significant innovation in our field.
>
> Thus, while the feature fusion in the backbone and decoder is a component of our model, the breadth and depth of our contributions, particularly in multi-modal exploration and lane-centric clustering, are what we believe to be the major points of novelty and impact in our work.

---

> ### Author Response · Authors · 2023-11-20
> **Response to Reviewer Z8i9 [3/3]**
>
> **Q1: *Regarding comparison with GroupLane.***
>
> We compare our method against the state-of-the-art model LATR. It is unfair to ask us to compare with GroupLane, which is under review at ICLR2024, the same as us. More critically, based on the results from GroupLane, when utilizing the same ResNet50 backbone, GroupLane exhibits inferior performance than LATR. Therefore, we would like to claim that our comparison is fair and competitive.
>
> **Q2: *Regarding using deformable attention.***
>
> We acknowledge the reviewer's concern regarding the innovation aspect of using deformable attention for aggregating multi-modal features.
> It is important to clarify that:
>
> **1)** Our contribution does not reside in using deformable attention per se, a well-established method widely employed in multi-modal/multi-view feature fusion.
>
> **2)** Instead, we propose the 3D DV Deformable Attention, an extension and adaptation of the original deformable attention.
>
> In Section 3.3, we meticulously outline the architecture and functionality of our 3D DV Deformable Attention, accompanied by Algorithm 2 and Figure 5 for clarity.
> Besides, we conduct a thorough comparison of our method with other methods that are also built upon the original deformable attention, showcasing the effectiveness of our approach. This comprehensive evaluation is summarized in Table 5, providing a nuanced understanding of our method's performance relative to existing approaches.

---

> > ### Author Response · Authors · 2023-11-23
> > **Further Feedback from The Reviewer**
> >
> > We thank the reviewer again for the valuable feedback and are happy to address any remaining concerns.

---

### Official Review · Reviewer_ECVn · 2023-10-31

**Soundness:** 3 good
**Presentation:** 3 good
**Contribution:** 3 good
**Rating:** 6
**Confidence:** 5

**Summary:**

This paper presents a novel 3D lane detection algorithm that exploits a dual-view representation. The proposed algorithm consists of bi-directional feature fusion to aggregate view-specific features, unified query generation that focuses on coherent lane features, and 3D dual-view deformable attention to associate information across the viewpoints. The authors provided outperformed experimental results on OpenLane dataset.

**Strengths:**

- Achieved the best performance on a 3D lane detection benchmark
- Leveraged multi-modal features to generate a unified query for 3D lane detection

**Weaknesses:**

- It appears that the 3D DV deformable attention mechanism lifts PV features to 3D using known camera parameters. Also, when the authors concatenate DV features, they depend on these camera parameters. Have the authors attempted to test the tolerance of calibration parameters to noise?
- Since the evaluation was only performed on OpenLane, it is difficult to check the generalizability of the proposed method.

**Questions:**

.

---

> ### Author Response · Authors · 2023-11-20
> **Response to Reviewer ECVn**
>
> We thank the reviewer for their positive feedback and address their concerns as below:
>
> **W1: *Tolerance of calibration parameters to noise.***
>
> Thank you for your insightful comment. In response, we conducted experiments to assess the impact of noise on the calibration parameters.
>
> #### **Noise Settings:**
> Following the methodology of Yu et al. (2023) [1], we introduced two noise settings: '**Noise**' and '**Stronger Noise (SNoise)**', which include rotation and translation randomly. In the 'Noise' setting, the rotation angle ranges from 1 to 5 degrees, and the translation ranges from 0.5cm to 1.0cm. In the 'Stronger Noise' setting, these values are doubled for a more pronounced impact.
>
> #### **Results without training noise:**
> We first test our model, which has **not** been exposed to any added noise during its training phase, under various noisy conditions.
> The results, presented in the first row of the table below, show a notable performance decline as noise intensity increases.
> Specifically, under a 0.7 probability setting, the performance deteriorates from 63.5 to 32.4/31.4 in the 'Noise'/'Stronger Noise' settings. For subsequent analyses, we designate this experiment, utilizing our original model, as the baseline for reference.
>
> #### **Enhancing robustness via training:**
> To enhance robustness, we incorporated calibration noise during the training phase. This strategy substantially reduces the performance degradation caused by noisy calibration, as shown in the second and third rows of the table below.
>
> #### **Comparative analysis:**
> When compared to the baseline (first row), we can observe that training with calibration noise significantly strengthens our model's robustness. It effectively maintains comparable results in noisy calibration conditions.
> The model trained with 'Stronger Noise' exhibits greater robustness than that trained with less intense noise, underscoring the benefits of this training strategy.
>
> Note: each result in the cell consists of   *F1-score / Accuracy*.
> | **Train @noise (N/SN)\|**  |  **Eval**  | **+ Noise**      |  |         |   **+ SNoise**           |      |   |
> |------|----------------|--------------|-----------------|--------------|--------------|--------------|--------------|
> |     | **Prob=0.0** | **Prob=0.3** | **Prob=0.5** | **Prob=0.7** | **Prob=0.3** | **Prob=0.5** | **Prob=0.7** |
> | -     @ Prob=0.0      | 63.5 / 92.4 | 52.2 / 89.9 | 40.9 / 85.6 | 32.4 / 82.5 |  52.0 / 89.0 | 40.3 / 83.4 | 31.4 / 79.2 |
> | **N** @ Prob=0.3   | 63.0 / 93.1 | 62.5 / 92.9 | 62.0 / 92.9  | 61.5 / 92.9  | 62.2 / 92.9 | 61.5 / 92.9 | 60.8 / 92.7 |
> | **SN** @ Prob=0.3  | 63.4 / 92.5 | 62.8 / 92.4 | 62.3 / 92.3    | 61.8 / 92.2 | 62.7 / 92.4 | 62.1 / 92.2 | 61.7 / 92.2 |
>
> We have included these robustness experiments in our updated Appendix A.3 for a comprehensive understanding, with a reference in purple font at the beginning of the main text on page 8. We appreciate your insightful feedback, which has been instrumental in strengthening our work.
>
> **W2: Generalizability**
>
> We appreciate your concern regarding the generalizability of our proposed method. OpenLane, being the first real-world and large-scale 3D lane dataset built upon Waymo, stands as a unique and valuable resource. The coupling of LiDAR data with OpenLane further enhances its significance.
>
> Currently, our evaluation is limited to this distinctive dataset due to its pioneering nature and the accessibility of coupled LiDAR data. While we acknowledge the importance of assessing generalizability across diverse datasets, the current focus on OpenLane provides a comprehensive exploration of our multi-modal method within the constraints of available resources. We plan to extend our evaluation to additional datasets as they become available and feasible for our study.
>
> We appreciate your understanding and consideration.
>
>
> ---
> Reference
>
> [1] Yu, Kaicheng, et al. "Benchmarking the robustness of lidar-camera fusion for 3d object detection." CVPR 2023.

---

> > ### Comment · Reviewer_ECVn · 2023-11-23
> >
> > I appreciate the authors' feedback and the extra efforts put into robustness experiments.

---

### Official Review · Reviewer_7Rt7 · 2023-11-06

**Soundness:** 3 good
**Presentation:** 4 excellent
**Contribution:** 3 good
**Rating:** 6
**Confidence:** 5

**Summary:**

This paper target 3D lane detection problem by proposing a dual view representation method utilizing both camera and Lidar as input. Their proposed method has achieve remarkable improvement over prior work with high efficiency.

**Strengths:**

1. Compared to prior work that mostly use only camera and feature learning in either Perspective View or Bird Eye View, this work fill the gap by fully investigate feature interaction of both sensor(Lidar+Camera) in both view(PV+BEV). And shows the superiority of their method by showing large improvement over prior work.
2. The work is very well written, clearly introduce all related work as well as their problems, and how this work target those problem by proposing new solutions.
3. This work include detail ablation study, make it clear to see the improvement over each step.

**Weaknesses:**

1. There’re some inconsistent in the evaluation result: 1). Although this method have explicitly filling the gap of missing Lidar modality in prior work, it might not seem as an apple to apple comparison in the table. In table 1 1.5m, compare with prior SOTA LATR, the proposed method didn’t achieve overwhelming improvement despite using one extra modality(Lidar). Does that mean the improvement in this range had been saturated to some extent? Also DV-3Dlane use a weaker backbone(ResNet-18/34) compare to ResNet50 LATR, make it even harder to see the full potential of proposed method compare to prior art. 2). Why Use OpenLane1000 in table 1(main experiment) but OpenLane300 for the ablation? Make it incomparable with result in Table1.
2. For onboard application, it might not always viable, to access both Lidar and Camera information for joint feature, does this method also provide implications for camera only/Lidar only methods? Maybe it’s more clear in ablation study if we could compare Table1 and Table2 directly. A follow up question, if camera-only ablation is possible, could it also show result on Apollo dataset? This is the dataset most commonly used in prior work.
3. For Figure1, please also consider adding more methods to be more completed, for instance light weight model like GenLane.
4. What is the ‘lane-aware prior knowledge’ in the abstract? I feel like ‘prior knowledge’ it is not discussed in the main text. Please explain or consider rephrase it.

**Questions:**

Please refer to weakness. In general I feel like this paper is well written, happy to raise my score if question addressed.
Some related papers from prior venue is not properly cited, please consider citing the following works

@inproceedings{liu2022learning,
  title={Learning to predict 3d lane shape and camera pose from a single image via geometry constraints},
  author={Liu, Ruijin and Chen, Dapeng and Liu, Tie and Xiong, Zhiliang and Yuan, Zejian},
  booktitle={Proceedings of the AAAI Conference on Artificial Intelligence},
  volume={36},
  number={2},
  pages={1765--1772},
  year={2022}
}

@inproceedings{yao2023sparse,
  title={Sparse Point Guided 3D Lane Detection},
  author={Yao, Chengtang and Yu, Lidong and Wu, Yuwei and Jia, Yunde},
  booktitle={Proceedings of the IEEE/CVF International Conference on Computer Vision},
  pages={8363--8372},
  year={2023}
}

@inproceedings{li2022reconstruct,
  title={Reconstruct from top view: A 3d lane detection approach based on geometry structure prior},
  author={Li, Chenguang and Shi, Jia and Wang, Ya and Cheng, Guangliang},
  booktitle={Proceedings of the IEEE/CVF Conference on Computer Vision and Pattern Recognition},
  pages={4370--4379},
  year={2022}
}

@inproceedings{ai2023ws,
  title={WS-3D-Lane: Weakly Supervised 3D Lane Detection With 2D Lane Labels},
  author={Ai, Jianyong and Ding, Wenbo and Zhao, Jiuhua and Zhong, Jiachen},
  booktitle={2023 IEEE International Conference on Robotics and Automation (ICRA)},
  pages={5595--5601},
  year={2023},
  organization={IEEE}
}

---

> ### Author Response · Authors · 2023-11-20
> **Response to Reviewer 7Rt7 [1/3]**
>
> We thank the reviewer for recognizing that our method fills the gap of missing LiDAR modality in prior work.
>
> **W1-1:*Regarding performance improvement and saturation under 1.5m setting***
>
> We appreciate the reviewer's observation and would like to clarify the evaluation results. First, we'd like to highlight the substantial error reduction by our method in both X and Z near/far ranges, showcasing impressive **46.12\%/48.26\%** and **57.33\%/49.04\%** improvements, respectively (see Table.1 of the main paper).
>
> While the F1 score may not fully capture the extent of this improvement, especially given the leniency of the 1.5m evaluation threshold, we believe there's room for further optimization, such as exploring more effective lane line representations.
>
> Meanwhile, to push beyond the results under the current lenient threshold, we propose to adopt a more strict threshold, 0.5m distance tolerance, for evaluation. This would offer a more nuanced examination of the model's accuracy, with the aim of encouraging further optimization and enhancing performance in alignment with considerations for real-world driving safety.
>
> Further, to address your concern regarding comparison, we integrated the SOTA method **LATR** with **LiDAR** inputs, projecting all points into the image and using them as additional features in the network. As shown in the following table, the results clearly demonstrate the positive impact of LiDAR, although it falls below the performance achieved by our proposed method. This comparison underscores the unique contribution of our work.
>
> It's worth noting that in the prior study, M$^2$-3DLaneNet, several multi-modal fusion approaches, such as PointPillar and BEVFusion, were compared, which exhibit subpar performance in the 3D lane task compared to M$^2$-3DLaneNet, falling behind our proposed method.
>
> | Model        | Backbone | F1   | Acc. | X error        | Z error        |
> |--------------|----------|------|------|----------------|----------------|
> | LATR         | Res50    | 54.0 | 91.7 | 0.171 / 0.201  | 0.072 / 0.099  |
> | LATR+LiDAR   | Res50    | 57.4 | 92.1 | 0.167 / 0.185  | 0.071 / 0.088  |
> | DV-3DLane    | Res34    | 63.5 | 92.4 | 0.090 / 0.102  | 0.031 / 0.053  |
> | DV-3DLane    | Res50    | 65.2 | 93.4 | 0.082 / 0.101  | 0.028 / 0.048  |
>
> **W1-2: *Regarding the backbone choice***
>
> Thank you for your thoughtful feedback. We appreciate your observation regarding the backbone choice. Our intention was to design an efficient model, leading us to adopt a smaller backbone. However, in response to your concern, we conducted experiments with ResNet-50, demonstrating that a larger backbone can further enhance model performance. The results are presented in the table referenced in **W1-1**. We hope this clarifies our design choice and addresses your comment effectively.
>
> **W1-3: *Regarding using OpenLane300 for the ablation***
>
> We follow previous works [1] [2] [3] to conduct ablation studies on the subset.
>
> ---
> Reference
>
> [1] Chen, Li, et al. "Persformer: 3d lane detection via perspective transformer and the openlane benchmark." ECCV 2022.
>
> [2] Huang, Shaofei, et al. "Anchor3dlane: Learning to regress 3d anchors for monocular 3d lane detection." CVPR 2023.
>
> [3] Luo, Yueru, et al. "LATR: 3D Lane Detection from Monocular Images with Transformer." ICCV 2023.

---

> ### Author Response · Authors · 2023-11-20
> **Response to Reviewer 7Rt7 [2/3]**
>
> **W2-1:*Regarding performance on OpenLane-1000 and implications of Camera-only/LiDAR-only methods.***
>
> We appreciate the reviewer's insightful query about the performance implications of our method with Camera-only and LiDAR-only modalities on the OpenLane-1000 dataset. To provide a comprehensive answer, we conducted two sets of experiments: one with single modality and single view, and another with single modality but dual views.
>
> | Line | Inputs     | View        | Backbone     | F1   | Acc. | X error        | Z error        |
> |------|------------|-------------|------------|------|------|----------------|----------------|
> | \#1  | Image      | PV          |Res34                | 52.9 | 90.3 | 0.173 / 0.212  | 0.069 / 0.098  |
> | \#2  | LiDAR      | BEV         |PillarNet34            | 54.1 | 84.4 | 0.282 / 0.191  | 0.096 / 0.124  |
> | \#3  | Image      | Dual Views  |Res34  | 54.3 | 91.5 | 0.165 / 0.200  | 0.067 / 0.094  |
> | \#4  | LiDAR      | Dual Views  |PillarNet34 | 55.3 | 87.9 | 0.156 / 0.143  | 0.031 / 0.050  |
> | \#5  | Image + LiDAR  | Dual Views  |Res34 + PillarNet34 | 63.5 | 92.4 | 0.090 / 0.102  | 0.031 / 0.053  |
>
> In the first experiment, we evaluated single-modality and **single-view** models, as detailed in Table.2 of the main paper, now applied to the OpenLane-1000 dataset. The outcomes, shown in rows #1 and #2 of the table above, present the performance using individual modalities.
>
> In the second experiment, we delved into single-modality but **dual-view** approaches:
> - For the image-based model, we adopted a strategy akin to BEVFormer, utilizing deformable attention to transform image features into BEV features. Then, we apply our dual-view decoder upon this, and the outcomes are illustrated in **row #3** of the table above.
> - For the LiDAR-based approach, we projected LiDAR point cloud features onto the 2D image plane to generate perspective-view features. The results of this approach are presented in **row #4** of the table above.
>
> Our results, as demonstrated in the table above, underscore that the dual-view representation significantly enhances the performance of baseline models in single-modal scenarios (comparing #1 with #3 and #2 with #4). This improvement confirms the effectiveness of our dual-view approach in learning 3D lane detection. Most notably, the combination of image and LiDAR modalities, coupled with our dual-view representation, achieves the best results, as shown in **row #5**. This synergy of modalities underlines the superiority of our proposed method.
>
> **W2-2: *Regarding camera-only performance on the Apollo dataset.***
>
> To address your concern, we conduct our image-only model on Apollo following the Table 2 setting of our main paper.
>
> |     Scene       | Backbone     |   F1   |   AP   | X error    | Z error    |
> |-----------------|---------------|-------|-------|------------|------------|
> | Balanced Scene  | Res34    |  96.4 | 97.6  | 0.046 / 0.299 | 0.016 / 0.213 |
> | Rare Subset     | Res34         |  95.6 | 97.2 | 0.071 / 0.664 | 0.025 / 0.568 |
> | Visual Variations | Res34       | 91.3 |  93.4 | 0.095 / 0.417 | 0.040 / 0.320 |
>
> **W3: *Including more methods in Figure.1***
>
> We appreciate the reviewer's valuable suggestion regarding enhancing Figure 1 by including additional methods.
>
> Our current results for GenLaneNet and 3DLaneNet are referenced from prior works, like PersFormer [1], Anchor3DLane [2]. Unfortunately, the pretrained models for these methods are not yet publicly available, which hinders our ability to directly evaluate their performance under the 0.5m setting. We will attempt to reproduce these results and assess them in the 0.5m setting. We kindly request your understanding in this matter, given the time constraints of this discussion period.
>
>
> ---
> Reference
>
> [1] Chen, Li, et al. "Persformer: 3d lane detection via perspective transformer and the openlane benchmark." ECCV 2022.
>
> [2] Huang, Shaofei, et al. "Anchor3dlane: Learning to regress 3d anchors for monocular 3d lane detection." CVPR 2023.

---

> > ### Author Response · Authors · 2023-11-20
> > **Response to Reviewer 7Rt7 [3/3]**
> >
> > **W4: *Regarding the ``lane-aware prior knowledge" in the abstract.***
> >
> > Thank you for bringing attention to this matter. In the abstract, we introduced the concept of  ``lane-aware prior knowledge" in the context of our proposed unified query generation approach.
> >
> > To provide more clarity, our method involves generating lane-aware queries in both Perspective View (PV) and Bird's Eye View (BEV) spaces using instance activation maps (IAMs). These IAMs contribute rich semantic and spatial priors concerning lanes, forming what we term as 'lane-aware prior knowledge'. Further, we introduce a lane-centric clustering method to group these queries from both spaces, which consequently synergizes the lane-aware prior knowledge into the unified (clustered) queries.
> >
> > In fact, we delve into detailing the derivation of 'lane-aware' queries in the initial and subsequent sections of Sec3.2, titled **Dual-view Query Generation**, within the main paper. Further, we elaborate on the process of unified query generation based on the afore-obtained lane-aware queries in the subsequent section, denoted as **Dual-view Query Clustering**.
> >
> > To address this concern explicitly, we have revised and highlighted the explanation of 'lane-aware prior knowledge' in **red font** in the updated manuscript. We appreciate your diligence in reviewing our work and welcome any additional feedback or suggestions you may have upon reviewing the updated version.
> >
> > **Q1: *Missed citation.***
> >
> > Thank you for highlighting the omission of citations for related works.
> > We have thoroughly addressed this concern by updating the manuscript with the additions highlighted in red font. Please kindly refer to the updated manuscript for the appropriate citations.

---

> ### Comment · Reviewer_7Rt7 · 2023-11-23
>
> Thanks author for the detail reply and additional experiment provided.
>
> W3: GenLane have their open source code on github and the result should be pretty easy to reproduce.
>
> W4: I still find the term 'prior knowledge' confusing, work [1] and [2] shows example of adopting 'prior' in lane formation. What you describe is more like 'lane aware query' or 'lane aware anchor', than 'prior knowledge'. If you still want to use the word 'prior', you should provide explanation comparing prior method also adopting 'prior'.
> [1]Reconstruct from top view: A 3d lane detection approach based on geometry structure prior
> [2]Learning to predict 3d lane shape and camera pose from a single image via geometry constraints
>
> Author should consider putting results on other dataset, like Apollo into the main paper with detailed comparison to prior method, as this is one of the earliest and most popular 3d lane detection dataset. Instead, you can put detail ablation to appendix.
> Beside, author should modify the paper as promised above to reduce misunderstanding in mismatch in experiment settings. And integrate all changed above to your paper.
>
> Other than that, I am satisfied with the current work. I have changed my rating to 6.

---

> ### Author Response · Authors · 2023-11-23
>
> We appreciate the feedback from the reviewer. Below, we address the reviewer’s concerns:
>
> **W3:** We thank the thoroughness of the reviewer. GenLaneNet is indeed accessible on GitHub, based on Apollo. We have attempted to adapt it to the OpenLane dataset but didn’t achieve the results reported in prior work. Therefore, for the sake of precision in our comparison, we haven't included GenLaneNet in the figure under the 0.5m setting.
> Nonetheless, we are still attempting to reproduce the results and aim to evaluate them under the 0.5m setting. We appreciate your understanding and would love to share any additional insights gained from these efforts.
>
> **W4-1: *Regarding to 'prior knowledge'***
>
> We acknowledge the potential misunderstanding, and to ensure clarity, we have modified our manuscript to consistently use the term "lane-aware knowledge", **without using 'prior'**,  to avoid any confusion with the previously used "geometry prior"[1] or "geometry constraints"[2].
>
> These revisions have been incorporated into our updated manuscript, and we appreciate your diligence in bringing this concern to our attention. Please refer to the revised version for further clarification.
>
> **Results on Apollo:**
>
> We appreciate the valuable suggestion from the reviewer. It's important to note that the Apollo dataset is exclusively an Image-only dataset without LiDAR information, while our work specifically focuses on multi-modal 3D lane detection.
>
> Further, given the constraints of space in the main paper and the need for a comprehensive table to present results on **Apollo**, we have opted to include the detailed experimental results in **Appendix A.6**.  This ensures maintaining an integrated presentation in the main body of the paper. We believe this approach allows for a more in-depth examination without compromising the main narrative. we appreciate your understanding.
>
> **Singel-modality & Dual-view Results on OpenLane-1000:**
>
> We appreciate the valuable suggestion from the reviewer.
>
> - We have integrated the **single-modality results** (Image-Only / LiDAR-Only) based on OpenLane-1000, **LATR + LiDAR** results, and the results of **our ResNet50-based model** (now termed *DV-3DLane-Large*) into the main results in the updated main text (See Table.1). Our previous ResNet34-based model is now termed as *"DV-3DLane-Base"* and ResNet18-based model is now termed as *"DV-3DLane-Tiny"* in the Table.1 of the manuscripts.
>
> Correspondingly, the performance of DV-3DLane-(Tiny / Base / Large) are all included in the Figure.1 & Appendix various scenario study table.
>
> Further,  we have moved the bidirectional feature fusion ablation study into our Appendix A.5.
>
> All the above changes have been included in our revised version, *highlighted in red*. Please kindly refer to our updated manuscripts.
>
> ---
> [1] Reconstruct from top view: A 3d lane detection approach based on geometry structure prior.
>
> [2] Learning to predict 3d lane shape and camera pose from a single image via geometry constraints.

---

### Official Review · Reviewer_nYY6 · 2023-11-12

**Soundness:** 3 good
**Presentation:** 3 good
**Contribution:** 3 good
**Rating:** 6
**Confidence:** 3

**Summary:**

This paper presents a LiDAR Camera 3D-Lane detection model. This model consists of 4 main blocks: 1) two backbones (image in camera view (CV) and LiDAR (BEV) in bird-eye-view) linked by a bidirectional feature fusion, 2) a dual view query generation with clustering between BEV and CV queries, 3) a decoder with query clustering that produce point queries and 4) a 3D dual-view deformable attention producing a 3D lane prediction. The experimental part shows that the proposed model outperforms SOTA, including the last ICCV2023 papers.

**Strengths:**

One original contribution is the bidirectional feature fusion module used to train the backbones in both BEV and CV leveraging for each view information from both sensors. The experimental part shows that this mechanism increases the F1 score about 2%.
Another important part of the pipeline is a Unified Query Generation process that win less than 1% F1 score.
The authors a 3D Dual-view Deformable Attention model that slightly improves the F1 score of the model.
Regarding the ablation study, the sensor fusion win more than 10% regarding the LiDAR only and about 20% regarding only the camera.
Experiments have been achieved on the public dataset OpenLane. The proposed model outperforms SOTA.

**Weaknesses:**

there are no really weakness in the paper.
I should be interesting to experiment your models on other datasets like  OpenLane-Huawei: a dataset with more horizontal lines.

minor typo :

instruction <-> introduction
Increase or remove line (d) of fig 7. (figures are too small)

**Questions:**

Did you try to experiment your models on other datasets like  OpenLane-Huawei: a dataset with more horizontal lines?

---

> ### Author Response · Authors · 2023-11-20
> **Response to Reviewer nYY6**
>
> **W1: Typo and Presentation**
>
> Thank you for bringing the minor typo and visualization to our attention.
> - We have fixed the typo, changing "Instruction" to "Introduction."
> - We have addressed the concern regarding the size of row (d) in Figure 7 by making improvements to the plot. Moreover, We have included corresponding plot improvements in the Appendix. Additionally, a zoom-in reminder has been incorporated into the captions, highlighted in *brown font* for the convenience of your review (*Please kindly refer to the updated manuscript*).
>
> We believe these adjustments effectively address the raised concerns.  Your thorough review is greatly appreciated.
>
> **Q1:**
> We thank you for your positive evaluation of our work and appreciate your suggestion to test our models on the OpenLane-Huawei dataset, which features more horizontal lines. We acknowledge the potential benefits of such an experiment; however, there are specific reasons we have not yet applied our method to this dataset during this discussion period.
>
> First, the OpenLane-Huawei dataset, primarily designed for topology reasoning among centerlines and traffic elements, is based on multi-view images. Our current methodology is based on the single-view setting, and adapting it to multi-view contexts is non-trivial. We recognize the importance of this adaptation and plan to explore it in future studies.
>
> Second, the OpenLane-Huawei dataset comprises two subsets: `SetA` and `SetB`. SetB, sourced from Nuscenes, Z values of centerline annotations are set to 0s built upon HD map. In this work, we focus on 3D space lane detection, making SetB unsuitable for our current approach.
>
> Regarding SetA, sourced from Argoverse-v2, it does have valid z-coordinate information. Nonetheless, the OpenLane-Huawei dataset, centering on the vision-centric approach, does not officially provide LiDAR data. While it's possible to obtain this data from the Argoverse repository, the large scale of the dataset posed a challenge for us to download and integrate within our work during the limited discussion period.
>
> In light of these considerations, we were unable to apply our method to the OpenLane-Huawei dataset during this discussion period. We sincerely appreciate your understanding in this regard. Moving forward, we plan to extend our method to incorporate both its camera and LiDAR data and will ensure to share our findings, as well as release the source code.
>
> Thank you again for your insightful suggestions and support for our work.

---

> > ### Comment · Reviewer_nYY6 · 2023-11-23
> >
> > Thanks to the authors for the detail reply. I appreciate this feedback and the answers given to my questions and concerns.

---

### Author Response · Authors · 2023-11-23
**Overall Response to the Reviewers**

Dear Reviewers,

We sincerely appreciate the time and effort you dedicated to reviewing our work.
We have carefully considered and addressed each of the individual concerns and feedback provided by each reviewer in our respective replies.
We thank all the constructive comments from your end, which strengthen our work. We have updated our manuscripts according to the reviewers' suggestions and uploaded them. Please kindly refer to the updated files.

Furthermore, we would like to inform you that modifications suggested by Reviewer 7Rt7 have been incorporated into our revised version. We believe these updates enhance the overall quality and clarity of our submission.

Thank you once again for your valuable feedback and thoughtful insights. Your contributions have significantly contributed to the refinement of our work. Further, we are happy to address any remaining concerns.

Best regards,

The Authors.

---

### Meta-Review · Area_Chair_ds45 · 2023-12-15

**Metareview:**

A 3D-Lane detection model for LiDAR cameras is presented in this paper. The four primary components of this model are as follows: 1) a bidirectional feature fusion links the two backbones (image in camera view (CV) and LiDAR (BEV) in bird-eye-view); 2) a dual view query generation process clusters BEV and CV queries; 3) a decoder that generates point queries through query clustering; and 4) a 3D dual-view deformable attention that generates a 3D lane prediction. The experimental section demonstrates that the suggested model beats SOTA on the OpenLane dataset, which includes the most recent ICCV2023 papers.

## Strengths

The study introduces a bidirectional feature fusion module for training backbones in both BEV and CV, increasing the F1 score by 2%. It also includes a Unified Query Generation process and a 3D Dual-view Deformable Attention model. The sensor fusion model outperforms SOTA in ablation studies, with over 10% success in LiDAR and 20% in camera. The study is well-written, addressing related work and proposing new solutions. It achieved the best performance on a 3D lane detection benchmark.

## Weaknesses

• Inconsistent results: Despite using Lidar modality, the proposed method didn't achieve significant improvement compared to prior SOTA LATR.
• Weak backbone: DV-3Dlane uses a weaker backbone (ResNet-18/34) compared to ResNet50 LATR, making it difficult to see the full potential of the proposed method.
• Use of OpenLane1000 instead of OpenLane300 for ablation: This could make it incomparable with the results in Table1.
• Questions: Does this method also provide implications for camera-only/Lidar-only methods?
• Recommendations: Add more methods to Figure1, like a lightweight model like GenLane.
• Lack of discussion on 'lane-aware prior knowledge': The 'lane-aware prior knowledge' is not discussed in the main text.
• Dependence on camera parameters: The 3D DV deformable attention mechanism lifts PV features to 3D using known camera parameters.
• Difficulty in checking the method's generalizability: The evaluation was only performed on OpenLane.

**Justification For Why Not Higher Score:**

As mentioned below, the reviewers are in favor of accepting the paper as a poster.

**Justification For Why Not Lower Score:**

Most of the reviewers highlighted  the recommendation of accepting the paper although it depends on the other submissions in the conference. The two extreme cases are not included on the final decision.

---

### Decision · Program_Chairs · 2024-01-16

Accept (poster)